# CTRLsum: Towards Generic Controllable Text Summarization

## Abstract

Current summarization systems yield generic summaries that are disconnected from users' preferences and expectations. To address this limitation, we present CTRLsum, a novel framework for controllable summarization. Our approach enables users to control multiple aspects of generated summaries by interacting with the summarization system through textual input in the form of a set of keywords or descriptive prompts. Using a single unified model, CTRLsum is able to achieve a broad scope of summary manipulation at inference time without requiring additional human annotations or pre-defining a set of control aspects during training. We quantitatively demonstrate the effectiveness of our approach on three domains of summarization datasets and five control aspects: 1) entity-centric and 2) length-controllable summarization, 3) contribution summarization on scientific papers, 4) invention purpose summarization on patent filings, and 5) question-guided summarization on news articles in a reading comprehension setting. Moreover, when used in a standard, uncontrolled summarization setting, CTRLsum achieves state-of-the-art results on the CNN/DailyMail dataset.[1]

## 1 Introduction

Neural summarization systems aim to compress a document into a short paragraph or sentence while preserving key information. There are largely two categories of summarization systems: extractive summarization that extracts important portions of a document (Cheng & Lapata, 2016; Nallapati et al., 2017; Narayan et al., 2018), and abstractive summarization that freely generates novel sentences (Rush et al., 2015; See et al., 2017; Paulus et al., 2018) which can produce coherent and fluent summaries more flexibly. In this paper we focus on abstractive summarization.

Typically abstractive summarization methods take a document as input and yield a generic summary to cover certain information identified by the model. However, content of interest is user-dependent. Summaries should select information with respect to preferences of a user. For example, Figure 1 shows an NBA basketball news article, and the reference summary describes several match results. However, fans of certain basketball stars in these teams such as Lebron James or Stephen Curry might only be interested in the matches they played and would like to know the player's scores as well.

Motivated by this, we focus on controllable summarization which allows the users to manipulate the summaries from the model. We propose CTRLsum, a framework to control summaries through *control tokens* in the form of a set of keywords or descriptive prompts. At training time, the model learns to predict summaries conditioned on both the source document and keywords that serve as external guidance. During inference, keywords and optional prompts, which are the target prefix to constrain decoding, are combined as control tokens to convey user preferences as shown in Figure 1.

Keywords and prompts are complementary. Prompts do not perform well in many cases such as entity or length controlled summarization as our preliminary experiments imply, but keywords can achieve those goals in a flexible way, for example, by using entity as keywords or varying the number of keywords to control entities and length respectively. However, keywords struggle in more open-ended scenarios like summarizing a list of contributions of scientific papers, while constraining the decoding with prompt "`the main contributions of this paper are:(1)`" is possibly sufficient to achieve the goal.

---

[1]Code and model checkpoints will be public after the review period.

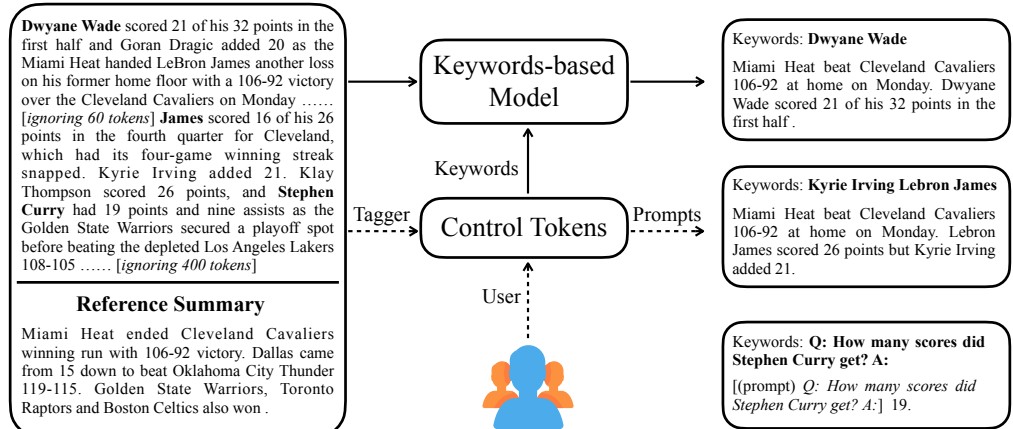

Figure 1: Workflow of the CTRLsum framework at inference time. Users interact with summaries through textual control tokens in the form of keywords or prompts. Keywords are required as input during training and testing, while prompts are optionally used at test time. Dashed lines represent optional paths – control tokens can come from the source article, user, or both. The right portion of the figure shows actual outputs from CTRLsum.

CTRLsum is trained using only keywords as additional input which can be easily identified from training summaries. It requires neither extra human annotations nor pre-defining control aspects for training, yet is quite flexible to achieve a broad scope of text manipulation as we will show in this paper. In contrast, prior work primarily rely on pre-defined "control codes" (Fan et al., 2018; Liu et al., 2018; Keskar et al., 2019), thus need to collect annotations for training and cannot generalize to unseen control aspects easily at test time.

We use pretrained BART (Lewis et al., 2019) as the underlying architecture and perform experiments on three datasets in three distinct domains: CNN/Dailymail news articles (Hermann et al., 2015), arXiv scientific papers (Cohan et al., 2018), and BIGPATENT patent documents (Sharma et al., 2019). We quantitatively evaluate CTRLsum on five control aspects: (1) entity-centric (§4.2) and (2) length-controllable summarization (§4.3), (3) summarizing the contributions of scientific papers, (4) summarizing the purpose of an invention (§4.4), and (5) summarizing answers to given questions in a zero-shot reading comprehension setting (§4.5). Notably, our approach also achieves comparable or superior performance to the strong BART summarization model on all datasets in a standard, uncontrolled setting (§4.6), leading to state-of-the-art results on the CNN/Dailymail dataset.

## 2 CTRLSUM

### 2.1 OVERVIEW

Unconstrained neural summarization methods are trained to learn the conditional distribution $p(\mathbf{y}|\mathbf{x})$, where $\mathbf{x}$ and $\mathbf{y}$ represent the source document and summary respectively. The generated summaries depend solely on the document $\mathbf{x}$ without human involvement. To control the output summaries, we propose using additional control tokens $\mathbf{z}$ to represent user preferences and training a summarization model that predicts the conditional distribution $p(\mathbf{y}|\mathbf{x}, \mathbf{z})$.

The control tokens $\mathbf{z}$ include keywords as extra inputs during training and inference. They can also optionally include prompts at test time to further constrain the decoding process. As shown in Figure 1, control tokens – in the form of keywords, prompts, or a combination of both – act as an interface between users and an otherwise black-box neural model, providing a flexible way for users to explicitly control automatic summarization. Next we describe how to obtain automatic keywords for training as well as potential applications at test time.

### 2.2 AUTOMATIC KEYWORD EXTRACTION

In addition to extracting keywords from training data to train the model, CTRLsum also features an automatic keywords extraction mechanism at test time, which can be used to suggest automatic

keywords according to user preferences, or perform uncontrolled summarization without user signals. Next we describe the keywords extraction methods at training and inference time respectively.

**Training.** For training, we use the ground-truth summary to identify keywords in the source document. Specifically, we first greedily select sentences from the document that maximize the ROUGE scores (Lin, 2004) with the reference summary. This step constrains keywords to those found in important sentences. Then, we identify all the longest sub-sequences in the extracted sentences that have matched sub-sequences in the ground-truth summary, similar to the copying word recognition method in (Gehrmann et al., 2018). Finally, we remove duplicate words and stop words and keep the remaining tokens as keywords. Compared to other keywords extraction methods (Riloff & Lehnert, 1994; Mihalcea & Tarau, 2004) which output only a few salient words, our extraction retains most content words found in the summary. This encourages dependence on the given keywords by building a reliable correlation between their presence in the input and the target. It in turn ensures that user-provided keywords are not ignored by the model at test time, which is catastrophic for a controllable summarization system.

**Inference.** We formulate the keyword extraction problem at test time as a sequence labeling task. Concretely, we train a BERT-based sequence tagger (Devlin et al., 2018) on the keywords and documents from training dataset. This tagger then computes the selection probability $q_j$ for each token in the test document. Similar to training time extraction, we first select $n_s$ sentences with the highest average token selection probability. Within these sentences words with $q_j > \epsilon$ are selected as keywords up to a maximum number of $m_{\max}$. The three hyperparameters $n_s, \epsilon, m_{\max}$ are selected based on the uncontrolled summarization performance on validation datasets. The results are reasonably robust to different settings (see Appendix D for details).

## 2.3 Summarization: Training Details

**Format.** At training time we prepend the keyword sequence to the source document separated with a special token. The summarization model is then trained to maximize $p(\mathbf{y}|\mathbf{x}, \mathbf{z})$ in an end-to-end fashion. The keyword sequence maintains the order of the keywords as they were in the source document, but we observe that the model often ignores this ordering as it frequently differs between source and target summary. We also separate keywords from different source sentences with the special token ("|"). In applications where the sentence boundary is unknown, as when users propose their own keywords, the "|" token can be ignored as in some of our experiments.

**Keyword Dropout.** As mentioned in §2.2, our keyword extraction strategy retains most words from the summary found in the source document. Without regularization, the dependence on such keywords is strong enough that the model rarely generates novel words in the summary. To remedy this, we randomly drop keywords at training time so that the model learns to rely on keywords that are present in the input, while also learning to still carry over key information from the source document that is not present in the keywords. Note that keywords dropout is applied at training time only.

Next we are going to introduce the five control aspects that we study in this paper as example use cases of CTRLsum. Qualitative examples of them are shown in Table 1.

## 2.4 Summarization: Inference with Keywords.

The keywords provide a generic interface to control multiple aspects of summaries, which allows the user to optionally rely on automatically extracted keywords, user provided keywords, or a combination of both. This method provides clean separation of test-time user control and the training process, including pretraining. Consequently, CTRLsum can be adapted to new use cases without changing model parameters. For example, though nothing during training specifically focuses on controlling entities or length, examples below demonstrate the general applicability of keyword control to entity and length manipulation.

**Entity Control.** The goal of entity control is to produce summaries that focus on entities of interest. Figure 1 exemplifies summarization with respect to different players when those player names are included as keywords directly influencing the summary.

**Length Control.** Users may have different preferences as to the length of summaries. We allow such manipulation of the summary length through a user-specified length parameter. Specifically, we

Table 1: Qualitative examples from the output of CTRLsum. Left column shows source or the generic reference summary. Keywords are bolded. "[]" denote that the tokens are used as both keywords and prompts.

| Source or Reference | Control Aspect | Keywords (bolded) or Prompts and Model Output |
|---|---|---|
| *Source:* Hundreds of additional Iraqi troops are being sent to reinforce colleagues who are trying to fend off ISIS' attempt to overrun Iraq's largest oil refinery, a key paramilitary force said Tuesday. The reinforcements come four days after ISIS began attacking northern Iraq's Baiji oil refinery, a key strategic resource that has long been a target because the facility refines much of the fuel used by Iraqis domestically. The additional troops came from Camp Speicher, a fortified Iraqi base near the city of Tikrit, according to the media office of the Hasd Al-Shaabi militia. The reinforcements include two federal police regiments, an Iraqi military quick reaction force battalion and a regiment from Hasd Al-Shaabi. [ignoring 110 tokens] The refinery is 40 kilometers (25 miles) from Tikrit. | Entity | **ISIS** – The reinforcements come four days after ISIS began attacking Baiji oil refinery. 

 **Hasd Al-Shaabi** – The reinforcements come from Camp Speicher, a fortified Iraqi base near Tikrit. They include two federal police regiments, an Iraqi military quick reaction force battalion and a regiment from Hasd Al-Shaabi. |
| | Length | (Length bucket 0) **Iraqi troops ISIS oil refinery \| come days attacking Baiji refinery \| base Tikrit** – The reinforcements come four days after ISIS began attacking Iraq's Baiji oil refinery. The additional troops came from Camp Speicher, a fortified Iraqi base near Tikrit. |
| | QA | **[Q: Where did the additional troops come from? A:]** – Camp Speicher near the city of Tikrit. |
| *Reference:* multiple sclerosis is a degenerative condition of the central nervous system that affects nearly 2.5 million of individuals in terms of their [ignoring 60 tokens]. in this work we resort to regularized machine learning methods for binary classification and multiple output regression we propose a pipeline that can be used to predict the disease progression from patient reported measures . the obtained model is tested on a data set collected from an ongoing clinical research project . | Contribution | **[the main contributions of this paper are : ( 1 )]** – we propose a regularization - based data analysis pipeline for the prediction of the transition point between relapsing - remitting and secondary - progressive forms of multiple sclerosis ; ( 2 ) we test the performance of the proposed pipeline on a data set collected from patients currently enrolled in an ongoing funded project ; ( 3 ) we discuss the potential role of patient reported outcome measures ( _ proms _ ) in the study of the disease evolution . |
| *Reference:* an implantable intraocular pressure sensor system has a sealed geometric shape with an internal pressure at a first value . the sealed geometric shape has a first light permitting surface and a second flexible surface . a pair of photocells is located in the sealed geometric shape . a light shield is coupled to the second flexible surface . when the second flexible surface is deflected , a light measurement by the pair of photocells indicates an intraocular pressure condition . | Purpose | **[the purpose of the present invention is]** – to provide an intraocular pressure measuring system that is small enough to be implanted in the eye of a patient |

first separate the training data into 5 buckets by summary length so that each bucket has the same number of examples. Then we compute the average number of keywords $K_l$ for each bucket on the training data. At test time, a user can specify length parameter $l \in \{0, 1, 2, 3, 4\}$ to include the $K_l$ keywords with the highest selection probability computed by the sequence tagger. This is similar to (Saito et al., 2020a), which uses the number of "guiding words" to control summary length.

## 2.5 SUMMARIZATION: INFERENCE WITH KEYWORDS AND PROMPTS

Prompts are pre-defined text sequences used as the target prefix to constrain decoding. They have been utilized to perform multi-purpose text generation with a single unified model (Radford et al., 2019; Brown et al., 2020). In the CTRLsum framework, prompts are a kind of control token sequence, and we always use such tokens as *both* the target prefix and keywords (ablation results on using prompts as keywords or prefix alone can be found in Appendix C). We find that using prompts as keywords besides prefix helps focus on prompt-related content and mitigate the over-generation issue of vanilla summarization models, as we will show in §4.4. To the best of our knowledge, we are the first to evaluate such a prompt-based control method for summarization systems.

**Summarizing Contributions.** Existing datasets about scientific papers such as arXiv (Cohan et al., 2018) collect paper abstracts as the summaries, which often include extra background context and lack detailed contribution descriptions for the associated paper. In many cases, readers would benefit from an explicit list of contributions in order to understand the novelty and value of the paper. For these cases, we propose using control tokens – `the main contributions of this paper are:(1)`. This prompt then triggers generation of a summary focused on contributions.

**Summarizing Invention Purpose.** Patent article summaries in existing datasets such as BIG-PATENT (Sharma et al., 2019) can be over-complicated, often covering core method details. Yet for a non-technical reader it would be preferred to provide a one-sentence summary that states the purpose of the invention while ignoring technical details. To apply CTRLsum in this scenario, we use the

control tokens, "`the purpose of the present invention is`". This triggers a concise summary focused on patent purpose.

**Question-guided summarization.** Human summarization can be constrained by questions (Kryściński et al., 2019) that require answers to be found in the summary. This points to an important connection between summarization and reading comprehension that we further explore. We hypothesize that a summarization model can directly answer some questions about the article if guided properly. This suggests the possibility of subsuming reading comprehension as a form of summarization. To verify this hypothesis, we use the control tokens "`Q: question text? A:`" to trigger reading comprehension behaviour.

We note that prompts- and keywords-based control are complementary in practice – while prompts could theoretically achieve any type of control, empirically they often do not work well for many aspects and the model is very sensitive to the precise wording of the prompt. For example, we found that using prompts such as "`a summary focused on [entity] is:`" or "`a short summary is:`" does not work as well as explicitly using keywords for entity or length control (details can be found in Appendix C).

## 3 RELATED WORK

Previous work on controllable summarization often collects control codes such as entity or length as supervision to train the model conditioned on both the code and article together (Fan et al., 2018; Liu et al., 2018). These methods do not generalize for controlling aspects of the summarization that were not seen during training. Recently Saito et al. (2020a) use the number of word prototypes to control summary length in a similar way to how we use keywords. Interactive summarization provides a way for users to continuously control the information that is included in the summary (Bornstein et al., 1999; Leuski et al., 2003). More broadly, controllable text generation has been studied for styles (Hu et al., 2017; Fu et al., 2018; He et al., 2020b), topics (Tang et al., 2019; Huang et al., 2019), and templates (Guu et al., 2018; Wiseman et al., 2018; He et al., 2020a).

Keyword-guided text generation has been applied in other contexts with different motivations. Gehrmann et al. (2018) utilize copying words at test time to mask copying operations in a summarization task. Li et al. (2018) and Saito et al. (2020b) use keywords as extra input to improve the uncontrolled summarization performance. Wang et al. (2016), Mou et al. (2016), and Yao et al. (2019) use textual input to plan poetry, dialogue, and stories respectively. Lexically-constrained decoding specifies certain lexicons as hard constraints in the target text (Hokamp & Liu, 2017; Post & Vilar, 2018). Prefix-constrained decoding was used in machine translation (Knowles & Koehn, 2016; Wuebker et al., 2016) and also to demonstrate the multi-task ability present in large pretrained models (McCann et al., 2018; Radford et al., 2019; Keskar et al., 2019; Brown et al., 2020).

## 4 EXPERIMENTS

Our experiments below are designed to (1) test the control efficacy of CTRLsum on five different aspects, and (2) examine the performance of CTRLsum in a traditional summarization setting without external control signals. Also, extensive model output examples can be found in Appendix E.

### 4.1 EXPERIMENTAL DETAILS

We perform experiments on three distinct-domain summarization datasets: CNN/Dailymail (CNNDM) news articles (Hermann et al., 2015), arXiv scientific papers (Cohan et al., 2018), and BIGPATENT patent articles (Sharma et al., 2019). For all datasets the source documents are truncated to 1024 tokens and the target summaries are truncated to 256 tokens following (Zhang et al., 2019). The conditional distribution $p(\mathbf{y}|\mathbf{x}, \mathbf{z})$ in CTRLsum is our fine-tuned version of the pretrained BART$_{\text{LARGE}}$ model (Lewis et al., 2019), which achieves state-of-the-art performance on several summarization benchmarks. The automatic keyword tagger at test time is based on the pretrained BERT$_{\text{LARGE}}$ model (Devlin et al., 2018) fine-tuned as described in §2.2. Our summarization model implementation is based on the fairseq toolkit (Ott et al., 2019) and the automatic keyword extraction model is based on the HuggingFace Transformers library (Wolf et al., 2019). Complete setup and training details can be found in Appendix A.1.

Table 2: Summarization performance with oracle entity or length signals from the reference summary. "CTRL-sum (automatic)" represents our model using automatic keywords in an uncontrolled setting. LengthCode is a length-control baseline. Both BART and LengthCode numbers are from our runs.

| Model | CNNDM | | arXiv | |
| --- | --- | --- | --- | --- |
| | ROUGE-1/2/L | BERTScore | ROUGE-1/2/L | BERTScore |
| BART (Lewis et al., 2019) | 44.24/21.25/41.06 | 0.336 | 45.16/17.36/40.55 | 0.164 |
| CTRLsum (automatic) | 45.65/22.35/42.50 | 0.363 | 46.91/18.02/42.14 | 0.169 |
| LengthCode (Fan et al., 2018) | 43.44/21.10/40.35 | 0.346 | 45.91/17.33/41.38 | 0.147 |
| CTRLsum (oracle entity) | **48.75/25.98/45.42** | **0.422** | – | – |
| CTRLsum (oracle length) | 46.26/22.60/43.10 | 0.365 | **47.58/18.33/42.79** | **0.173** |

Table 3: Entity control results on CNNDM. Success rate is the fraction of decoded summaries that actually mention the given entity, while factual correctness is the fraction of summaries that are judged as factually correct by human annotators. The BART numbers are in terms of unconstrained generated summaries. EntityCode numbers are directly from (Fan et al., 2018), which is obtained with a weaker convolutional seq2seq architecture and requires entity annotations at training time.

| Model | Success Rate (%) | | Factual Correctness | |
| --- | --- | --- | --- | --- |
| | Lead-3 | Full-article | Important | Unimportant |
| BART (Lewis et al., 2019) | 61.4 | 29.0 | 98.0 | – |
| EntityCode (Fan et al., 2018) | 61.2 | 33.8 | – | – |
| CTRLsum | **97.6** | **94.8** | **99.0** | 100.0 |

For evaluation, we measure commonly used ROUGE scores (Lin, 2004) and the recently proposed BERTScore (Zhang et al., 2020) when ground-truth is available. For control-related evaluation where we often do not have reference summaries, we (1) collect ground-truth summaries when possible, (2) examine whether summaries respect the control signal, or (3) resort to human evaluation.

## 4.2 ENTITY CONTROL

**Setup.** We first simulate user preference by providing the model with oracle entities extracted from the ground-truth target. Then we compare it to the model using automatic keywords in a uncontrolled setting to show the effect of oracle entities. To examine whether the decoded summaries respect entity change, we sample 100 documents and repeatedly acquire every entity in the document to generate summaries, following Fan et al. (2018). Then we compute *Success Rate*, the fraction of requested entity actually occurring in the output summaries. The results are reported in separation of whether the entity is from leading 3 sentences or from the full article. To test if the summaries from different entity input are factually consistent with the document, we sample another 100 documents, and for each we randomly sample one "important" entity that appears in the reference, and one "unimportant" entity that occurs neither in the reference nor the leading three source sentences to produce summaries. For each (article, summary) pair we ask 3 annotators from Amazon Mechanical Turk to make a binary decision as to whether the summary can be entailed from the article. We then take the majority vote as the result and report the fraction of factually correct summaries. We evaluate on CNNDM only since many examples in arXiv and BIGPATENT do not have identifiable entities.

**Results.** In Table 2 we observe that the use of oracle entities helps boost the ROUGE-2 score by 3.6 points compared with using automatic keywords, which means CTRLsum is able to take advantage of the given entities. Table 3 shows the Success Rate and factual correctness evaluations. We include the numbers from Fan et al. (2018) (EntityCode) for reference point. We note that their numbers come from a convolutional seq2seq architecture (see Appendix B for ablation analysis on this) and their method utilizes entity annotations during training time, thus is not very comparable to CTRLsum. Remarkably, our model achieves a high success rate for both lead-3 and full-article entities reaching around 95%. Yet other systems struggle to include the given entities especially for the ones that do not occur in the beginning of the article. Factual correctness scores from human annotators suggest that CTRLsum is able to generate factually consistent summaries no matter whether the entity of interest is important or not, comparable to the unconstrained BART baseline.

Table 4: Length control performance. MAD measures the deviation of output length from reference length, while PCC represents the correlation between given length signal and the actual output length.

Table 5: F1 scores on the dev set of NewsQA and SQuAD. GPT2 results are from our runs. The BART baseline and GPT2 use prompts while CTRLsum use the same trigger as both keywords and prompts.

| Model | CNNDM | | arXiv | |
|---|---|---|---|---|
| | MAD ↓ | PCC ↑ | MAD ↓ | PCC ↑ |
| BART | 1.20 | 0.00 | 1.08 | 0.00 |
| CTRLsum (automatic) | 1.25 | 0.00 | 0.98 | 0.00 |
| LengthCode (Fan et al., 2018) | 1.17 | -0.02 | 1.06 | 0.00 |
| CTRLsum (+length) | **0.87** | **0.53** | **0.69** | **0.48** |

| Model | NewsQA | SQuAD v1.1 |
|---|---|---|
| **Supervised** | | |
| SpanBERT (Joshi et al., 2020) | 73.0 | 94.6 |
| MatchLSTM (Wang & Jiang, 2017) | 49.6 | 70.0 |
| **Zero-Shot** | | |
| GPT2-Large (774M params, w/o fine-tuning) | 24.9 | 23.5 |
| BART (406M params, w/o fine-tuning) | 8.2 | 15.8 |
| BART (406M params, fine-tuned on CNNDM) | 32.6 | 41.7 |
| CTRLsum (406M params, trained on CNNDM) | **48.2** | **59.6** |

## 4.3 LENGTH CONTROL

**Setup.** Similar to entity control, we first examine the effect of oracle length signal from the reference to simulate user preference. In addition to ROUGE and BERTScore, we measure the length distance between the decoded summary and the reference following (Liu et al., 2018). Specifically, we compute the mean of absolute deviation (MAD) of the actual length bucket code $l_{\text{sys}}$ of the decoded summary from the ground-truth control code $l_{\text{ref}}$, as $\frac{1}{N} \sum_{n}^{N} |l_{\text{sys}}^{(n)} - l_{\text{ref}}^{(n)}|$. To assess the summary variations as length signals change, we further sample 1000 documents and decode 5 different-length summaries for each document. Then we report the Pearson Correlation Coefficient (PCC) between the input bucket code and actual bucket code. Experiments are conducted on CNNDM and arXiv.

**Results.** In Table 2 CTRLsum with oracle length signals only presents relatively small gains over the automatic CTRLsum baseline. This implies that oracle lengths only convey limited additional information to help generate the reference summary. We also run the LengthCode baseline (Fan et al., 2018) based on BART, where the ground-truth length bucket code is prepended to the article at both training at test time. However, LengthCode fails to consistently improve over BART with oracle length signals. Moreover, we find that the BART model fine-tuned with LengthCode method almost ignores the length signal with PCC close to 0, as shown in Table 4. This is not very surprising since length code would be less useful when the summarizers grow stronger, which can already learn a good length predictor implicitly. In contrast, CTRLsum with length-guided keywords achieves high positive PCC between control signal and actual output length, and is able to reduce the length deviation MAD compared to automatic baselines.

## 4.4 CONTRIBUTION AND PURPOSE SUMMARIZATION

**Contribution Summarization Setup.** There is no existing dataset to evaluate contribution summarization of scientific papers, bringing challenges to our evaluation. However, researchers often summarize the bullet contributions of their paper in the Introduction section, which inspire us to extract such contribution claims as the reference summary. Therefore, we resort to the entire arXiv database,[2] and download all the papers whose first submission time is within the first six months of 2019[3] that gives us 67K papers. We extract the Introduction section and bullet contributions with regular expression and filter out the ones that fail. The contributions are used as the reference and the Introduction section after removing the contribution claims is used as the source article – we aim to predict contributions from the rest of the introduction section. This procedure leads to 1018 test examples. We test the model trained on arXiv.

**Purpose Summarization Setup.** To collect a test dataset that features one-sentence invention purpose summaries, we sample 1000 test examples from BIGPATENT and present their reference summaries to human annotators from Amazon Mechanical Turk. For each example we ask one annotator to select the sentence that convey the purpose of the invention. We also provide the option for annotators that the invention purpose cannot be identified. After filtering out the invalid examples, we collect 763 examples as our test data.

---

[2]We do not use the arXiv test set because we can only extract 20 valid test points from it. The entire arXiv database is at: https://www.kaggle.com/Cornell-University/arxiv

[3]The arXiv dataset used to train CTRLsum is collected before April 2018 according to their paper submission time, thus there should be no data overlap between the training data and our contribution test data.

Table 6: Summarization performance on contributions of papers and purpose of inventions. The BART baseline uses prompts while CTRLsum use the same trigger as both keywords and prompts.

| Model | Contribution | | Patent Purpose | |
|---|---|---|---|---|
| | ROUGE-1/2/L | BERTScore (P/R/F1) | ROUGE-1/2/L | BERTScore (P/R/F1) |
| BART (prompt) | 43.84/17.46/25.89 | 0.119/0.142/0.130 | 29.05/**11.80**/22.50 | 0.016/0.236/0.107 |
| CTRLsum (prompt+keyword) | **43.88/18.17/27.79** | 0.179/0.098/**0.138** | **33.64**/11.37/**24.24** | 0.180/0.152/**0.165** |

Table 7: Uncontrolled summarization performance. Automatic keywords are from the sequence tagger, while oracle keywords are obtained utilizing the gold summaries. We report the oracle performance for a reference point. The BART results are from our runs. BS denotes BERTScore.

| Model | CNNDM | | arXiv | | BIGPATENT | |
|---|---|---|---|---|---|---|
| | ROUGE-1/2/L | BS | ROUGE-1/2/L | BS | ROUGE-1/2/L | BS |
| CTRLsum (Oracle Keywords) | 64.65/40.42/60.92 | 0.555 | 56.08/25.31/50.23 | 0.268 | 55.19/26.62/47.10 | 0.291 |
| BART (Lewis et al., 2019) | 44.24/21.25/41.06 | 0.336 | 45.16/17.36/40.55 | 0.164 | 45.83/19.53/39.47 | 0.187 |
| PEGASUS (Zhang et al., 2019) | 44.17/21.47/41.11 | – | 44.70/17.27/25.80 | – | **53.63/33.16/42.25** | – |
| CTRLsum (Automatic Keywords) | **45.65/22.35/42.50** | **0.363** | **46.91/18.02/42.14** | **0.169** | 45.80/18.68/39.06 | **0.188** |

**Results.** Table 6 shows results of contribution summarization on scientific papers and invention purpose summarization on patent filings. Through using the prompt text as both the decoder prefix and keywords, CTRLsum outperforms the BART baseline in most cases. We further report the precision (P) and recall (R) scores in BERTScore besides F1. We observe that the BART baseline tends to over-generate a full summary with low precision scores while CTRLsum is able to focus on keywords-related content.

## 4.5 QUESTION-GUIDED SUMMARIZATION

**Setup.** We directly test question-guided summarization on reading comprehension benchmarks in a zero-shot setting. Specifically, we evaluate the CNNDM summarization models on in-domain NewsQA (Trischler et al., 2017) and out-of-domain SQuAD 1.1 (Rajpurkar et al., 2016) respectively. We note that some NewsQA test articles are present in the CNNDM summarization training dataset, yet we think it is still a reasonable unsupervised setting since our model never sees questions or answers during training. In addition to comparing with the vanilla BART model, we also include the zero-shot performance from GPT2 language models (Radford et al., 2019) (without fine-tuning) as a reference point. We omit the largest GPT2 model with 1.5B parameters since it cannot be evaluated in our single GPU device due to memory limits. We report F1 scores on the two benchmarks.

**Results.** BART is pretrained with a denoising task to predict the denoised version of the source, and performs poorly on zero-shot reading comprehension out of box, as shown in Table 5. Interestingly, however, BART fine-tuned on a summarization task – without seeing any question-answer pairs in the training data – is able to improve the F1 scores by 24.4 and 25.9 points on NewsQA and SQuAD respectively. Moreover, CTRLsum equipped with question keywords is able to further boost the performance by 15.6 and 17.9 points, approaching the supervised MatchLSTM (Wang & Jiang, 2017) score on NewsQA. Such results suggest that summarization might be a suitable transfer task for abstractive reading comprehension, which we leave for future work to explore.

## 4.6 AUTOMATIC SUMMARIZATION

Table 7 shows the uncontrolled summarization performance without any user input, where our method uses the automatically extracted keywords as described in §2.2. On CNNDM and arXiv datasets CTRLsum outperforms the strong BART and PEGASUS baselines by a large margin, leading to new state-of-the-art performance on CNNDM. It also performs comparably to the BART baseline on BIGPATENT in terms of BERTScore, though with an inferior ROUGE-2 score. Yet there is a big performance gap between BART-based models and PEGASUS on BIGPATENT. The reasons might be different dataset processing,[4] sub-optimal learning schedule, or inherent difference between BART and PEGASUS.

Table 8: Human evaluation scores (scale 1-5, higher is better) on entity control and purpose control experiments. Control accuracy (CA) and control relevance (CR) are reported. A score significantly different (according to the Welch Two Sample t-test, with $p < 0.05$) than CTRLsum is denoted by $*$.

| Model | Important Entity | | Unimportant Entity | | Purpose | |
|---|---|---|---|---|---|---|
| | CA | CR | CA | CR | CA | CR |
| CTRLsum | 3.5 | 4.2 | 4.0 | 4.0 | 4.0 | 3.7 |
| BART | 3.8 | 3.7* | 1.3* | 1.2* | 4.0 | 3.0* |

Table 9: Human evaluation scores (scale 1-5, higher is better) of uncontrolled summarization performance. Evaluation Dimensions from left to right are: factual consistency (FAC), relevance (REL), fluency (FLU), coherence (COH). A score significantly different (according to the Welch Two Sample t-test, with $p < 0.05$) than CTRLsum (Automatic Keyword) is denoted by $*$.

| Model | CNNDM FAC/REL/FLU/COH | arXiv FAC/REL/FLU/COH | BIGPATENT FAC/REL/FLU/COH |
|---|---|---|---|
| CTRLsum (Automatic Keyword) | 4.6/4.6/4.1/4.1 | 4.1/4.3/4.1/4.1 | 4.2/4.2/4.0/4.1 |
| BART | 4.6/4.7/4.2/4.1 | 4.1/4.1*/3.9/4.0 | 4.2/4.3/4.1/4.0 |
| CTRLsum (Oracle Keyword) | 4.6/4.7/4.1/4.1 | 4.2/4.3/4.0/4.1 | 4.2/4.2/4.2*/4.1 |

## 4.7 HUMAN EVALUATION

In this section we present human evaluation results for both controlled and uncontrolled summarization. Full experiment details can be found in Appendix A.2.

**Controlled Summarization.** We present further human evaluation results to evaluate "control" directly by informing annotators the intended control signal. We conduct experiments on entity and purpose control. Specifically, we inform the annotators our intent (to obtain summaries focused on a specific entity or purpose of patent), then we ask them to provide scores in scale 1-5 over two dimensions: (1) Control Accuracy (CA): whether the summary contains accurate main information with respect to the intent, and (2) Control Relevance (CR): how the summary is relevant to the control intent overall – a summary that contains redundant contents that are unrelated to the intent will be penalized. Results including significance tests are shown in Table 8. The control accuracy for important entity control and purpose control are comparable between BART and CTRLsum without significant difference (p-value > 0.05), while CTRLsum shows significantly better control relevance overall by focusing on the desired information. Also, the unconstrained BART are unable to generate unimportant-entity-related summaries and thus suffers from poor scores on both dimensions.

**Uncontrolled Summarization.** We follow (Grusky et al., 2018; Fabbri et al., 2020) to ask human annotators from Amazon Mechanical Turk to score summaries (scale 1-5) over four dimensions: (1) Factual Consistency (FAC): the summary should only contain statements that can be entailed by the source document, (2) Relevance (REL): the summary should only contain *important* information of the source document, (3) Fluency (FLU): each sentence in the summary should be fluent, and (4) Coherence (COH): the summary should be well-structured and well-organized. Results including significance tests are present in Table 9. The quality of summaries from all systems on all dimensions is generally good with a score mostly higher than 4.0. However, most scores do not show significant difference from CTRLsum (Automatic Keyword) with large p-values, despite their very different similarities against the reference summaries in terms of ROUGE/BERTScore (e.g. CTRLsum with oracle keywords). This implies that the summary quality from different systems powered by strong pretrained models like BART has become difficult to be clearly distinguished by non-expert MTurkers. We also note that non-expert human judgement for summarization may be unreliable and exhibit poor correlation with expert judgement (Gillick & Liu, 2010; Fabbri et al., 2020).

## 5 CONCLUSION

In this paper we propose a generic framework to perform multi-aspect controllable summarization. The model is conditioned on keywords to predict summaries during training. At inference time the control tokens, in the form of keywords or prompts, enable users to interact with models in a very flexible way. Experiments on five different control aspects demonstrate the efficacy of our method.

---

[4]PEGASUS updated the BIGPATENT data to preserve casing and applied some format cleaning.

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

# A    EXPERIMENTAL SETUP DETAILS

## A.1    GENERAL SETUP

In this section we include additional experimental details left out in the main content due to space limitations. We fine-tune the pretrained BART$_{\text{LARGE}}$ model in all our experiments. Specifically we use the `bart.large` checkpoint from fairseq (Ott et al., 2019). For all BART-based summarization models, we fine-tune with learning rate 3e-5 and a polynomial learning rate decay schedule, the optimizer is Adam (Kingma & Ba, 2015) and batch size is 64. Our optimization scheme and hyperparameters follow the BART fine-tuning instructions in fairseq examples. We train the summarization models with 20k steps on CNNDM, 50k steps on arXiv, and 300k steps on BIGPATENT. We train the BERT tagger with learning rate 5e-5, Adam optimizer, and batch size of 128 on all datasets. Similar to summarization models, the tagger is trained with 20k, 50k, and 300k steps on CNNDM, arXiv, and BIGPATENT respectively. Also, we adopt a sliding window approach so that the BERT-based tagger is able to handle sequences that are longer than 512 tokens. For both ROUGE and BERTScore evaluation, we report the F1 measure. We report the rescaled BERTScore, and the hash code is `roberta-large_L17_no-idf_version=0.3.6(hug_trans=3.0.2)-rescaled`.

As mentioned in §2.2, we need three hyperparameters for automatic keywords extraction during inference – the number of pre-selected sentences $n_s$, the selection probability threshold $\epsilon$, and the maximum number of keywords $m_{\max}$. We select these hyperparameters for each dataset based on the uncontrolled summarization ROUGE-2 score on validation dataset. The summarization performance is robust to these hyperparameters in a reasonable range, as shown in Appendix D. Specifically, we use $\{n_s = 10, \epsilon = 0.25, m_{\max} = 30\}$ for CNNDM, $\{n_s = 10, \epsilon = 0.15, m_{\max} = 40\}$ for arXiv, and $\{n_s = 5, \epsilon = 0.15, m_{\max} = 30\}$.

**Invention Purpose Summarization.**    In the experiment of summarizing invention purpose on patent articles (§4.4). We examined whether the model would possibly copy source sentences through matching the prompts, we search strings in the form of "the purpose of [some words or phrases] is" among 763 test examples, and only 3 test articles are identified. This means the models are not generating by exactly matching prompts most of the time.

## A.2    HUMAN EVALUATION SETUP

Here we include details about human evaluation experiments in §4.7.

**Controlled Summarization.**    For controlled summarization, we sample 100 examples for each task, and summaries of each example from all systems are presented together to the human annotator to be scored. For CNNDM we provide article and summaries, while for BIGPATENT we provide reference and summaries using the reference summary as a surrogate for the source article. This is because the source patent documents are very long and hard to be read by non-expert humans. We did not evaluate contribution summarization since it is unrealistic to ask humans to judge contributions of many scientific papers from various domains. We tried to hire workers from Amazon Mechanical Turk first, but we failed to obtain reliable results from them – they often ignored the given user intent and tended to score the text as uncontrolled summaries (reflected by very poor scores on unimportant-entity summaries because these summaries do not contain the main information of the article), even though we instructed them that the control signal is critical. Therefore, we ask two independent human annotators through personal correspondence from the authors of this paper. One of the annotator is a PhD researcher on physics, and the other is a law graduate on intellectual property in the United States. They are able to follow the given control intent and considered more reliable than the MTurkers. We take the average of two annotators as the score for each example, and average over all examples to obtain the final score.

**Uncontrolled Summarization.**    For uncontrolled summarization, we sample 100 examples for each dataset, and hire 3 independent workers from Amazon Mechanical Turk to conduct evaluation. For CNNDM we provide article and summaries, while for arXiv and BIGPATENT we provide reference and summaries using the reference summary as a surrogate for the source article. This is because the source patent documents or scientific papers are very long and hard to be read by non-expert humans. Summaries of each example from all systems are presented together to the human annotator to be

scored. The median score of 3 workers is taken for each example, and average over all examples is reported.

## B    ABLATION ANALYSIS OF ENTITY CONTROL

In Table 3 we observe that CTRLsum achieves a very high success rate ($\sim 95\%$) of entity control, compared to previous work (Fan et al., 2018) which can only succeed 61.2% and 33.8% of the time on lead-3 and full-article entities respectively. We perform ablation analysis to understand the important gradients that contribute to the success of CTRLsum. We train CTRLsum with another two architectures in addition to BART: (1) convolutional seq2seq (Gehring et al., 2017) with the same hyperparameters as in (Fan et al., 2018), and (2) transformer seq2seq with the same hyperparameters as the base model in (Vaswani et al., 2017). Note that the transformer model is trained from scratch without pretraining. Results are shown in Table 10. CTRLsum parameterized with a weaker convolutional seq2seq architecture fails to depend on the keywords well with an over 40-point success rate drop, yet the success rate of transformer seq2seq without pretraining only drops around 5 points. This implies that the transformer seq2seq architecture is critical for CTRLsum to depend on the keywords well, while pretraining can further improves it.[5]

Table 10: Entity control results on CNNDM. Success rate is the fraction of decoded summaries that actually mention the given entity.

| Model | Success Rate (%) | |
| --- | --- | --- |
| | Lead-3 | Full-article |
| BART (Lewis et al., 2019) | 61.4 | 29.0 |
| Fan et al. (2018) | 61.2 | 33.8 |
| CTRLsum (Conv Seq2Seq) | 50.1 | 23.3 |
| CTRLsum (Transformer Seq2Seq) | 92.6 | 88.3 |
| CTRLsum (BART) | **97.6** | **94.8** |

## C    ABLATION ANALYSIS ON KEYWORDS AND PROMPTS

In the controlling aspects we studied CTRLsum uses control tokens either as keywords alone (entity and length), or as keywords and prompts together (contribution, purpose, QA). Here we present further results when control tokens are used as prompts, keywords, or both for entity control, contribution control, and NewsQA tasks. Specifically for entity control, we use the control tokens "`a summary focused on [entity] is:`" for "prompt" and "prompt + keyword" variants.[6] In this case success rate is computed excluding the prompt text. The control tokens for other settings are the same as previous experiments. Results are shown in Table 11, where keywords and prompts are of different importance for different tasks and are complementary in general. For example, using prompts to control entities turns out to be difficult with a very low success rate – we find that the system fails to understand the prompt and produce summaries appropriately in most cases. However, prompts contribute the most to contribution summarization with comparable performance with using prompts and keywords together, while removing prompts and using keywords alone suffers from drastic performance drop to trigger the contribution. For NewsQA task, prompts and keywords demonstrate mixing effectiveness – using either of them alone experiences over 20 F1 points loss compared to using them together.

---

[5]For reference points, the ROUGE-1/2/L scores (with automatic keywords) of CTRLsum (Conv Seq2Seq) is 41.19/18.71/38.05 while CTRLsum (Transformer Seq2Seq) obtained 43.69/20.78/40.55.

[6]We tried several prompts variants, for example, QA style ones "`Q: What happened to [entity]? A:`" or "`Q: What do we know about [entity]?  A:`". None of them lead to meaningful entity control.

Table 11: Ablation analysis on the role of keyword and prompt respectively. Entity success rate refers to the full article entity success rate.

| Model | Entity Success Rate (%) | Contribution ROUGE-1/2/L | BERTScore | NewsQA F1 |
|---|---|---|---|---|
| CTRLsum (keyword) | **94.8** | 39.96/12.74/22.68 | 0.088 | 15.5 |
| CTRLsum (prompt) | 17.6 | 43.82/18.12/27.64 | 0.133 | 26.3 |
| CTRLsum (prompt + keyword) | 12.6 | **43.88/18.17/27.79** | **0.138** | **48.2** |

# D   ROBUSTNESS ANALYSIS OF KEYWORDS EXTRACTION HYPERPARAMETERS

Table 12 shows the ROUGE-2 scores of uncontrolled summarization on the validation set with different keywords extraction hyperparameters. We use more fine-grained stride size to iterate the $m_{\max}$ hyperparameter for CNNDM since its source articles are usually shorter than arXiv and BIGPATENT. As observed, the automatic summarization performance is relatively robust to these hyperparameters in a reasonable range.

Table 12: ROUGE-2 scores of uncontrolled summarization on the validation set with different keywords extraction hyperparameters.

| Model | CNNDM | arXiv | BIGPATENT |
|---|---|---|---|
| $\epsilon = 0.10, n_s = 5, m_{\max} = 25$ | 22.82 | – | – |
| $\epsilon = 0.10, n_s = 5, m_{\max} = 30$ | 22.71 | 17.81 | 18.60 |
| $\epsilon = 0.10, n_s = 5, m_{\max} = 35$ | 22.54 | – | – |
| $\epsilon = 0.10, n_s = 5, m_{\max} = 40$ | – | 17.96 | 18.35 |
| $\epsilon = 0.10, n_s = 10, m_{\max} = 25$ | 22.83 | – | – |
| $\epsilon = 0.10, n_s = 10, m_{\max} = 30$ | 22.67 | 17.99 | 18.61 |
| $\epsilon = 0.10, n_s = 10, m_{\max} = 35$ | 22.44 | – | – |
| $\epsilon = 0.10, n_s = 10, m_{\max} = 40$ | – | 18.03 | 18.04 |
| $\epsilon = 0.15, n_s = 5, m_{\max} = 25$ | 22.85 | – | – |
| $\epsilon = 0.15, n_s = 5, m_{\max} = 30$ | 22.79 | 17.80 | **18.79** |
| $\epsilon = 0.15, n_s = 5, m_{\max} = 35$ | 22.71 | – | – |
| $\epsilon = 0.15, n_s = 5, m_{\max} = 40$ | – | 17.95 | 18.76 |
| $\epsilon = 0.15, n_s = 10, m_{\max} = 25$ | 22.85 | – | – |
| $\epsilon = 0.15, n_s = 10, m_{\max} = 30$ | 22.77 | 17.99 | 18.76 |
| $\epsilon = 0.15, n_s = 10, m_{\max} = 35$ | 22.41 | – | – |
| $\epsilon = 0.15, n_s = 10, m_{\max} = 40$ | – | **18.05** | 18.62 |
| $\epsilon = 0.20, n_s = 5, m_{\max} = 25$ | 22.86 | – | – |
| $\epsilon = 0.20, n_s = 5, m_{\max} = 30$ | 22.87 | 17.71 | 18.77 |
| $\epsilon = 0.20, n_s = 5, m_{\max} = 35$ | 22.89 | – | – |
| $\epsilon = 0.20, n_s = 5, m_{\max} = 40$ | – | 17.88 | 18.71 |
| $\epsilon = 0.20, n_s = 10, m_{\max} = 25$ | 22.87 | – | – |
| $\epsilon = 0.20, n_s = 10, m_{\max} = 30$ | 22.85 | 17.88 | 18.77 |
| $\epsilon = 0.20, n_s = 10, m_{\max} = 35$ | 22.84 | – | – |
| $\epsilon = 0.20, n_s = 10, m_{\max} = 40$ | – | 17.98 | 18.73 |
| $\epsilon = 0.25, n_s = 5, m_{\max} = 25$ | 22.84 | – | – |
| $\epsilon = 0.25, n_s = 5, m_{\max} = 30$ | 22.88 | 17.57 | 18.67 |
| $\epsilon = 0.25, n_s = 5, m_{\max} = 35$ | 22.91 | – | – |
| $\epsilon = 0.25, n_s = 5, m_{\max} = 40$ | – | 17.71 | 18.66 |
| $\epsilon = 0.25, n_s = 10, m_{\max} = 25$ | 22.90 | – | – |
| $\epsilon = 0.25, n_s = 10, m_{\max} = 30$ | **22.95** | 17.76 | 18.72 |
| $\epsilon = 0.25, n_s = 10, m_{\max} = 35$ | **22.95** | – | – |
| $\epsilon = 0.25, n_s = 10, m_{\max} = 40$ | – | 17.84 | 18.70 |
| $\epsilon = 0.30, n_s = 5, m_{\max} = 25$ | 22.58 | – | – |
| $\epsilon = 0.30, n_s = 5, m_{\max} = 30$ | 22.62 | 17.24 | 18.53 |
| $\epsilon = 0.30, n_s = 5, m_{\max} = 35$ | 22.63 | – | – |
| $\epsilon = 0.30, n_s = 5, m_{\max} = 40$ | – | 17.32 | 18.52 |
| $\epsilon = 0.30, n_s = 10, m_{\max} = 25$ | 22.65 | – | – |
| $\epsilon = 0.30, n_s = 10, m_{\max} = 30$ | 22.70 | 17.38 | 18.55 |
| $\epsilon = 0.30, n_s = 10, m_{\max} = 35$ | 22.70 | – | – |
| $\epsilon = 0.30, n_s = 10, m_{\max} = 40$ | – | 17.44 | 18.55 |

# E   RANDOM OUTPUT EXAMPLES

In this section, we randomly sample test examples and show the source aticle, reference summary, and the model output from CTRLsum for each control aspect.

## E.1   ENTITY CONTROL

For entity control, we randomly sample 3 articles from CNNDM and for each article we randomly select 5 entites as keywords to show the model output.

Table 13: Random Entity Control Examples

| | |
|---|---|
| Article | Americans on the United States' no-fly list will now be privy to information about why they have been banned from commercial flights and be given the opportunity to dispute their status, according to court documents filed by the Justice Department this week. The revised policy comes in response to a June ruling by a federal judge that said the old process was in violation of the Fifth Amendment's guarantee of due process. The decision was part of an American Civil Liberties Union lawsuit brought on behalf of 13 Americans on the list. But the ACLU isn't satisfied with the government's new policy, outlined in documents filed Monday in federal courts in Oregon (PDF) and Virginia (PDF). "After years of fighting in court for complete secrecy and losing, it's good that the government is finally now going to tell people of their status on the No Fly List," said Hina Shamsi, director of the ACLU National Security Project and the lead attorney on the case, in a statement. "Unfortunately, we've found that the government's new redress process falls far short of constitutional requirements because it denies our clients meaningful notice, evidence, and a hearing. The government had an opportunity to come up with a fair process but failed, so we're challenging it in court again." People on the no-fly list, managed by the FBI's Terrorist Screening Center, are prohibited from boarding a commercial flight for travel into or out of the United States. The number of people on the list is classified. An official with knowledge of the government's figures told CNN in 2012 that the list contained about 21,000 names, including about 500 Americans. Before the change, American citizens and permanent residents who inquired with the government about being denied aircraft boarding received a letter that neither confirmed nor denied their inclusion on the no-fly list. Now, they'll be made aware of their status if they apply for redress, with an option to request further information. "The U.S. government is making enhancements to the Department of Homeland Security Traveler Redress Inquiry Program (DHS TRIP) to provide additional transparency and process for U.S. citizens and lawful permanent residents who have been denied boarding on a commercial aircraft because they are on the No Fly List," the Department of Homeland Security said in a statement. In cases in which travelers included on the list request to receive or submit more information about their status, the government will provide a second, more detailed response, identifying "specific criterion under which the individual has been placed on the No Fly List," according to the court documents. An unclassified summary of that information will be provided "to the extent feasible, consistent with the national security and law enforcement interests at stake," court papers said. Those who appear on the no-fly list will then have further opportunity to dispute their status in writing, with supporting materials or exhibits, and will receive a final written decision from the Transportation Security Administration. The 2014 ruling that prompted the policy changes had called for passengers on the list to be given the opportunity to dispute their status before a judge. |
| Reference Summary | Americans on the no-fly list will now get info about why they've been banned from flights. ACLU says the policy still denies "meaningful notice, evidence, and a hearing" |
| CTRLsum | **the Fifth Amendment's**:  The new policy is in response to a June ruling that said the old process was in violation of the Fifth Amendment's guarantee of due process.

**Civil Liberties Union**:  The American Civil Liberties Union says it's not satisfied with the government's new policy.

**the Transportation Security Administration**:  The Transportation Security Administration will provide travelers with more information about their status. A June ruling had called for passengers on the list to be given the opportunity to dispute their status before a judge.

**Virginia**: The ACLU is challenging the new policy in federal courts in Oregon and Virginia.

**FBI**: People on the no-fly list, managed by the FBI's Terrorist Screening Center, are prohibited from boarding commercial flights. The ACLU isn't satisfied with the government's new policy. |

Table 14: Random Entity Control Examples

| | |
|---|---|
| Article | Point guard Stephen Curry nearly single-handedly outscored New Orleans with 11 first-quarter points as the Warriors built a 15-point lead and rolled to victory in Game One of their Western Conference first-round series. Game Two in the best-of-seven series is scheduled for Monday night in Oakland. Golden State, the top seed in the West, picked up right where it left off in the regular season, recording a 19th straight home win and 40th in 42 games this year. Stephen Curry scored a stunning 34 points for the Golden State Warriors in there play-off game. The Warriors did it by taking a 25-point lead into the final minute of the third quarter, then holding on. 'We missed a lot of free throws, which made it a lot closer than it needed to be,' coach Steve Kerr said. 'But in the playoffs you've just got to get it done somehow. We're up 1-0. That's where we want to be.' Curry led the Warriors with 34 points, hitting 13 of 25 shots and four three-pointers. All five Golden State starters scored in double figures. Guard Klay Thompson complemented Curry with 21 points, while power forward Draymond Green (15 points, 12 rebounds) and center Andrew Bogut (12 points, 14 rebounds) recorded double-doubles. The point guard has been in spectacular form as he looks to lead the Warriors to the NBA glory. Curry celebrates after scoring a three-pointer on his way to scoring 11 first quarter points. New Orleans power forward Anthony Davis scored a game-high 35 points, 20 in the fourth quarter. He also blocked four shots and grabbed seven rebounds. Small forward Quincy Pondexter, assigned to cover Curry most of the game, added 20 points for the Pelicans. . Visiting Washington outscored Toronto 11-4 in the overtime in winning the opener of their first-round playoff series. Washington traveled to Toronto and came away with the victory with a confident overtime performance. Forward Paul Pierce started the overtime with a three-pointer and scored 20 points to lead the Wizards, who let a 15-point lead slip away in the fourth quarter before winning. Guard Bradley Beal added 16 points for Washington and forward Nene contributed 12 points and 13 rebounds. Forward Amir Johnson came off the bench to lead the Raptors with 18 points and guard DeMar DeRozan added 15 points. . Paul Pierce (left) top-scored in the clash with 20 points as the Wizards secured an important play-off win. |
| Reference Summary | Stephen Curry scored 34 points for Golden State against New Orleans. The Californian-based team defeated the Pelicans 106-99. Washington Wizards outscored the Toronto Raptors 11-4 in overtime. Paul Pierce led the scoring with 20 points for the Wizards . |
| CTRLsum | **Stephen Curry**: Stephen Curry scored 34 points as Golden State Warriors beat New Orleans Pelicans. Washington Wizards beat Toronto Raptors in overtime in play-off game.

**Oakland**: Stephen Curry scored 34 points for the Golden State Warriors. Game Two in the best-of-seven series is scheduled for Monday night in Oakland.

**Steve Kerr**: Stephen Curry scored 34 points as the Warriors beat the Pelicans. Coach Steve Kerr said: 'We missed a lot of free throws, which made it a lot closer than it needed to be'

**Klay Thompson**: Stephen Curry scored 34 points as the Warriors beat the Pelicans. Klay Thompson and Draymond Green also scored in double figures .

**Andrew Bogut**: Stephen Curry scored 34 points as the Warriors beat the Pelicans. Draymond Green and Andrew Bogut both recorded double-doubles. |

Table 15: Random Entity Control Examples

| | |
|---|---|
| Article | It's the ultimate treat for Benedict Cumberbatch fans and stands an imposing 6ft tall - just like the man himself. But shoppers at London's Westfield Stratford City shopping centre looked more than a little surprised to discover a chocolate sculpture of Benedict Cumberbatch in their midst. One lady was spotted cautiously approaching the edible artwork before quickly backing off, while another couldn't quite hide their smile of surprise. Scroll down for video . Finishing touches: The sculpture is readied for its big unveiling at Westfield Stratford City shopping centre. Oh dear: Reaction to the sculpture was mixed, with some shoppers bursting into laughter. Even less impressed was the shopper who stood stony-faced in front of the creation for several moments, while another burst into laughter as soon as she spotted it. It did, however, prove an immediate hit with a pair of police sniffer dogs who wagged their tails as they gave it a thorough sniffing down. . The artwork, which has been given pride of place in the shopping mall's atrium, was commissioned by UKTV to mark celebrate its screening of the third series of Sherlock. It took a crew of eight people to complete the sculpture, which took over 250 man hours to create and weighs 40kg . Does it look like me? Benedict Cumberbatch strikes a pose with James Corden during an Oscars party. Mixed reaction: A pair of police sniffer dogs loved the sculpture but shoppers looked baffled. Hilarious: A lady bursts into laughter after spotting the 6ft homage to Mr Cumberbatch. Not amused: A shopper looks thoroughly unimpressed as she contemplates the artwork. Luckily for Cumberbatch, who usually enjoys a considerably more complimentary response to projects he's involved in, the piece will only be in residence temporarily. The 38-year-old actor, who is currently expecting his first child with wife Sophie Hunter, 37, isn't the only famous face to have found himself the subject of an edible artwork. . In the run up to the release of 50 Shades of Grey, bakers created not one but two 6ft gateaux paying homage to Jamie Dornan. One depicted the actor in the grey suit beloved of his 50 Shades character Christian Grey, while the other showed him topless and came complete with an edible six-pack. Award-winning: Both Jennifer Lawrence and her cake alter-ego have won awards. Homage: The cake, which triumphed at a show last November, was inspired by the Hunger Games . Actress Jennifer Lawrence has also been immortalised in cake, with baker Lara Clarke creating a sweet treat designed to resemble the 24-year-old's Hunger Games alter-ego, Katniss Everdeen. The confection, which was baked ahead of the release of Mockingjay Part One in November, met with the approval of Lawrence herself, who, when asked about it, said Ms Clarke was 'incredibly talented'. Other A-listers to get the culinary treatment include Kevin Bacon, whose likeness was made from bacon, and Taylor Swift who featured on a cake. The Duke and Duchess of Cambridge were turned into a tasty-looking pizza, while President Barack Obama had his likeness sculpted from butter. Immortalised: The Duke and Duchess of Cambridge have been turned into pizza in the past. ribute: Kevin Bacon has a bacon sculpture to his name while Jamie Dornan has been made into a cake. |
| Reference Summary | A 6ft chocolate sculpture of Benedict Cumberbatch has been unveiled. Toothsome statue has been placed inside a London shopping centre. But shoppers reactions to the creations were decidedly unenthusiastic. One woman glared at it while others just looked thoroughly baffled . It did manage to win the approval of pair of police sniffer dogs. It weighs 40kg and took eight people 250 man hours to create . Other celebrities to get culinary tributes include Jennifer Lawrence. Her 6ft cake won an award - and the 24-year-old's approval. Actor Kevin Bacon has also been immortalised - in bacon . |
| CTRLsum | **Westfield**: The sculpture was unveiled at London's Westfield Stratford City shopping centre.

**James Corden**: Benedict Cumberbatch is not the only famous face to be immortalised in chocolate. Jennifer Lawrence and James Corden have also been turned into cakes .

**Sophie Hunter**: Actor, 38, is expecting his first child with wife Sophie Hunter.

**Jamie Dornan**: Shoppers at London's Westfield Stratford City shopping centre looked baffled by the sculpture. Other famous faces to be immortalised in edible art include Jamie Dornan and Jennifer Lawrence.

**Hunger Games**: The sculpture was commissioned by UKTV to mark the screening of Sherlock. It follows in the footsteps of other A-listers such as Jamie Dornan and Jennifer Lawrence, who have been immortalised in cakes. Lawrence's Hunger Games cake won an award at a show last year . |

## E.2 PAIRED ENTITY CONTROL

The entity control experiments in this paper only consider one entity as the control signal, here we show examples inputting paired entities. Specifically, we are interested in the output when one of the paired entities is important and the other is unimportant. Therefore, we sample 3 articles from CNNDM and for each article we randomly select one important entity and one unimportant entity as paired keywords to show the model output. We repeat this sampling five times for each article to obtain five different summaries.

Table 16: Random Paired Entity Control Examples

| | |
|---|---|
| Article | A former U.S. Army enlistee who posted on Facebook about "the adrenaline rush" of dying in jihad was arrested Friday and charged with trying to detonate a car bomb at Fort Riley military base in Kansas, authorities said. A second man, who allegedly knew about the bomb plot but didn't call authorities, was charged with failing to report a felony. John T. Booker Jr. of Topeka, an American citizen also known as Mohammed Abdullah Hassan, was taken into custody near Manhattan, Kansas, in a van that contained what he thought was a bomb, the criminal complaint said. The "bomb" had actually been put together by two confidential informants with nonexplosive materials, the complaint said. Fort Riley's security was never breached and no people were in danger, the U.S. Justice Department said in a press release. Booker enlisted in the Army last year and was due to ship out to basic training April 7, 2014, said Army spokesman Wayne Hall. The criminal complaint said the FBI questioned him March 24, 2014 about comments posted on Facebook, such as, "Getting ready to be killed in jihad is a HUGE adrenaline rush. I am so nervous. NOT because I'm scare to die but I am eager to meet my lord." Booker waived his Miranda rights and told the agents he enlisted to commit an insider attack against American soldiers like Maj. Nidal Hassan had done at Fort Hood, Texas, the complaint said. Hassan opened fire in a building in November 2009, killing 13 people and wounding more than 30. His enlistment was terminated March 24, 2014, at the request of Army Criminal Investigation Command, Hall said. Booker began communicating with a confidential informant later in 2014, the complaint said, and often talked about his plans to engage in violent jihad in support of ISIS. He and the informant watched ISIS videos together, the complaint said, and Booker talked about how he wanted to go to Iraq and turn his weapon on American soldiers when ordered to shoot the enemy. On March 9, Booker said he believed ISIS wanted him to commit a truck bombing in the United States and thought a good target would be nearby Fort Riley, a large Army base that's home to the 1st Infantry Division, known as "The Big Red One." Booker said "that detonating a suicide bomb is his No. 1 aspiration because he couldn't be captured, all evidence would be destroyed and he would be guaranteed to hit his target," the criminal complaint said. He made a video with a Fort Riley airfield in the background and said ISIS was coming to kill American soldiers, both abroad and in the United States, the complaint said. Booker acquired components for a bomb and rented a storage locker to store the components, the complaint said. The plan was for confidential informants to build a bomb and for Booker to drive to Fort Riley and detonate it, the complaint said. But the bomb was built with "inert" parts and would never explode, the complaint said. On Friday, the informants and Booker drove to what Booker thought was a little-used utility gate near Fort Riley, the complaint said. While Booker was making final connections on the "bomb," the FBI arrested him, the complaint said. He was charged with one count of attempting to use a weapon of mass destruction, one count of attempting to damage property by means of an explosive and one count of attempting to provide material support to the Islamic State of Iraq, a designated foreign terrorist organization. If convicted, he could face life in prison. Alexander E. Blair, 28, of Topeka was taken into custody Friday and charged with failing to report a felony. The FBI said agents interviewed Blair after Booker's arrest. Blair said he shared some of Booker's views, knew of his plans to detonate a vehicle bomb at Fort Riley and loaned him money to rent storage space, according to the FBI's criminal complaint. He said he thought Booker would carry out his plan but did not contact authorities, the complaint said. If convicted, Blair faces a maximum of three years in prison. |
| Reference Summary | Alexander Blair, 28, of Topeka accused of knowing about bomb plot but not contacting authorities. Fort Riley's security was never breached and the device was "inert" and not a threat, authorities say. John T. Booker Jr., 20, of Topeka had acquired bomb parts and made a propaganda video, the Justice Department says. |
| CTRLsum | **Fort Riley ISIS**: John T. Booker Jr. is charged with trying to detonate a car bomb at Fort Riley. A second man, who allegedly knew about the bomb plot but didn't call authorities, is charged. Booker said he believed ISIS wanted him to commit a truck bombing in the United States. |
| | **John T. Booker Jr. FBI**: John T. Booker Jr. enlisted in the Army last year and was due to ship out to basic training April 7, 2014. The FBI questioned him March 24, 2014 about comments posted on Facebook. |
| | **John T. Booker Jr. Infantry Division**: John T. Booker Jr. enlisted in the Army last year and was due to ship out to basic training April 7, 2014. He planned to detonate a car bomb at Fort Riley, a large Army base that's home to the 1st Infantry Division. |
| | **Justice Department Islamic State of Iraq**: The Justice Department says Fort Riley's security was never breached and no people were in danger. John T. Booker Jr. is charged with attempting to use a weapon of mass destruction. He is also charged with trying to provide material support to the Islamic State of Iraq. |
| | **Alexander Blair ISIS**: John T. Booker Jr., 28, charged with trying to detonate a car bomb at Fort Riley. Alexander E. Blair, 28, is charged with failing to report a felony. Booker said he wanted to commit an insider attack against American soldiers in support of ISIS. |

Table 17: Random Paired Entity Control Examples

| | |
|---|---|
| Article | Ander Herrera has caught the eye in recent weeks after cementing a spot in Manchester United's starting line-up but it appears he does not actually have a clear sight at goal. Herrera netted a brace against Aston Villa on Saturday afternoon without actually looking at Brad Guzan's goal as his eyes appeared to be shut when making contact with the ball. In fact, six of Herrera's seven goals have been scored without him even having to glimpse at either the ball or the opposition's net. . Manchester United star Ander Herrera scores his side's opening goal against Aston Villa with his eyes shut. The Spanish midfielder appears to have his eyes closed as he strikes at Brad Guzan's goal . Aston Villa and England midfielder Fabian Delph attempts to block Herrera's left-footed shot. His eyes were wide open when he struck an impressive first-time shot against Yeovil in the third round of the FA Cup back in January. . However his double against Aston Villa and his goals against Queens Park Rangers, Leicester, Preston and Swansea all came without Manchester United's summer signing having to make eye contact with the ball. Herrera appears to have a history of shooting with his eyes closed as the image of his goal for Spain's Under 20 side back in 2009 shows. Herrera hits the back of the net while representing Spain's Under 20 side against Tahiti . The former Athletic Bilbao joined the Red Devils for £29million in the summer due to his vision and creativity in midfield. . Louis van Gaal heaped praise on the 25-year-old's shooting ability after Saturday's Premier League encounter at Old Trafford, by stating: 'He has a very good kicking technique and he should be more composed,' Van Gaal said to MUTV. 'I said to him 'you have to control the ball before you shoot'. 'I said that to him again in yesterday's training session, and then I kissed him at half-time because he had controlled the ball for the first time in his life and then he shot.' . Herrera's eyes were also shut for Manchester United's third goal against Aston Villa on Saturday. His only Manchester United goal scored with his eyes wide open came against Yeovil Town back in January . Herrera scored in Manchester United's 4-0 win over Queens Park Rangers without looking at the ball. The former Athletic Bilbao man flicked the ball past Preston's players without having his eyes on the ball . Herrera scored in Manchester United's 2-1 defeat by Swansea City at the Liberty Stadium in February. He also netted in Manchester United's shock 5-3 defeat at the hands of Premier League strugglers Leicester. |
| Reference Summary | Ander Herrera has scored seven goals for Man United since joining in June. Herrera's eyes have been shut when striking the ball for six of his goals. His superb strike against Yeovil Town has been only goal with eyes open. Herrera netted a brace in Manchester United's 3-1 win over Aston Villa . |
| CTRLsum | **Ander Herrera Van Gaal**: Ander Herrera has scored six of his seven goals without looking at the ball . Louis van Gaal has praised the Spanish midfielder's shooting ability . 

 **Yeovil Town Swansea City**: Ander Herrera scored his side's opening goal against Aston Villa with his eyes shut. The 25-year-old has also scored without looking at either the ball or the opposition's net in games against QPR, Leicester City, Swansea and Yeovil Town . 

 **Manchester United Queens Park Rangers**: Ander Herrera appeared to have his eyes closed as he scored against Aston Villa on Saturday. The 25-year-old has scored six of his seven goals without looking at the ball. Herrera also netted with his eyes open in Manchester United's 4-0 win over Queens Park Rangers . 

 **Aston Villa MUTV**: Ander Herrera scored a brace against Aston Villa on Saturday. But the Spanish midfielder appeared to have his eyes closed . Louis van Gaal told MUTV that Herrera should be 'more composed' . 

 **Herrera Athletic Bilbao**: Ander Herrera has scored six of his seven goals with his eyes closed. The former Athletic Bilbao man joined Manchester United for £29million in the summer . |

Table 18: Random Paired Entity Control Examples

| | |
|---|---|
| Article | A husband accused of trying to murder his cheating wife was ready to let her have sex with another man once a month as long as she stayed with him, his trial heard yesterday. A close friend told how Joseph O'Riordan, 74, had confided in him about the extraordinary plan for his 47-year-old wife Mandy. O'Riordan, a councillor and former nightclub owner, stabbed her eight times in a jealous rage after finding out she had been having an affair with a postman. . Extraordinary deal: Joseph O'Riordan stabbed his wife of ten years Amanda (left) with a seven inch kitchen knife eight times - yesterday Brighton Crown Court heard he was considering allowing her to have affairs. She suffered life-threatening injuries after being knifed in the torso, chest, arms and back. The jury was also shown dramatic footage of the moment police arrived at the couple's home to be greeted by a 'calm' O'Riordan opening the door. The revelation of his proposal for keeping his wife of ten years came from Alfred Harris. He told how O'Riordan had confided five days before the attack that he believed she was having an affair. O'Riordan was 'choked up and emotional' when he said: 'I think Amanda is playing away. She's getting her nails and hair done more regularly, she's been on a diet and doesn't want sex.'. Asking for a suit: O'Riordan sent his wife this letter from his prison cell. The following day, added Mr Harris, the men met for a pub lunch in O'Riordan's home village of Polegate, East Sussex. 'I saw Joe and he told me that Amanda had been seeing someone else – a guy who drove a van. Joe said he loved Amanda to bits and if she wanted to have sex with someone else once a month that would be okay as long as she stayed with him.'. In a statement read to Brighton Crown Court, Mr Harris also described the couple as 'loving and close'. . He was 'so shocked' to learn that O'Riordan had attacked his wife at their flat on a residential care home estate. The jury saw images of four police officers, one of whom was wearing a lapel camera, arriving shortly before 10pm last October 22 after racing to the scene. . PC Dave Catt said they drew their 'incapacitating' sprays fearing the knifeman would be still holding his weapon. They were greeted by O'Riordan wearing a blood-spattered light blue shirt and holding a cordless phone on which he had phoned for an ambulance. Mr Catt said O'Riordan admitted: 'I found out that she was having an affair and I lost it.'. Mrs O'Riordan was moaning and lying on a bed, holding a towel to her stomach with a deep chest wound and serious wounds to her hand and back. Paramedics arrived moments later and took her to hospital. Jurors looked at two screens as images of her husband's arrest and subsequent detention at a police station were shown. Growing suspicion: Giving evidence yesterday Alfred Harris – a friend of the couple for more than six years – told how O'Riordan had confided in him that he believed his wife was having an affair. Yesterday, jurors at Brighton Crown Court (above) were shown dramatic footage of the moment police arrived at the couple's home to be greeted by a 'calm' Mr O'Riordan opening the door. PC Stuart Kenway told how, as O'Riordan had opened the door, he 'appeared calm and composed and the situation was surreal' as he then said: 'She is in the bedroom – do you want the knife?' Officers were directed to a 7in blade with a black handle which was in the kitchen. Dr Stephen Drage, an intensive care consultant with Brighton and Sussex University Hospitals, told the jury how seriously Mrs O'Riordan was hurt. 'It is quite clear she was bleeding to death,' he said. 'She underwent life-saving surgery which took six hours.'. O'Riordan denies attempted murder. The trial continues. . . Sorry we are not currently accepting comments on this article. |
| Reference Summary | Joseph O'Riordan, 73, stabbed wife eight times after discovering her affair. She was left with life-threatening injuries to her torso, chest, arms and back. Yesterday Brighton Crown Court heard about deal he was ready to offer her. He had told friend about the idea while in the pub just days before stabbing. |
| CTRLsum | **Joseph O'Riordan Alfred Harris**: Joseph O'Riordan, 74, is accused of stabbing wife Mandy, 47, eight times. Friend Alfred Harris told how he had told him about the extraordinary plan. 

**Brighton Crown Court Stephen Drage**: Joseph O'Riordan, 74, accused of stabbing wife Mandy, 47, eight times. Brighton Crown Court heard he was considering allowing her to have affairs. Dr Stephen Drage, an intensive care consultant, told jury how she was 'clearly hurt' 

**Joseph O'Riordan Catt**: Joseph O'Riordan, 74, is accused of stabbing wife Mandy, 47, eight times. PC Dave Catt said he 'lost it' when he found out about the affair. 

**Stuart Kenway Joseph O'Riordan**: Joseph O'Riordan, 74, is accused of stabbing wife Mandy, 47, eight times. PC Stuart Kenway told how he 'appeared calm and composed' . 

**Brighton Crown Court Amanda**: Joseph O'Riordan, 74, is accused of stabbing wife Amanda, 47, eight times. He denies attempted murder at Brighton Crown Court. |

### E.3 Length Control

For length control, we randomly sample 3 test articles from CNNDM and for each article we generate five different-length summaries.

Table 19: Random Length Control Examples. Control tokens are bolded.

| | |
|---|---|
| Article | Film director David Lynch has confirmed he will no longer direct the revival of "Twin Peaks" – a cult 1990s television show that was set to return in 2016. The offbeat TV series, created by Lynch and Mark Frost, featured a quirky FBI agent who went to the Pacific Northwest town of Twin Peaks to investigate the mysterious murder of a high school girl named Laura Palmer. The groundbreaking series is considered one of the most influential shows in television history. Lynch broke the news about his departure in a series of tweets, saying that the show's third season will continue without him. He said he felt the network was not offering enough money to produce the show "the way it needed to be done." Lynch also wrote that he had personally called the actors over the weekend to let them know he would no longer be directing. Showtime Network, which will air the nine-episode comeback, released a statement saying they were "saddened" by Lynch's decision. "We were saddened to read David Lynch's statement today since we believed we were working towards solutions with David and his reps on the few remaining deal points," read the statement. "Showtime also loves the world of Twin Peaks and we continue to hold out hope that we can bring it back in all its glory with both of its extraordinary creators, David Lynch and Mark Frost, at its helm." Showtime announced they would produce a third season in October last year. Actor Kyle MacLachlan, who played the coffee-obsessed FBI agent Dale Cooper in the original series, had confirmed he would reprise the lead role for the new season. |
| Reference Summary | David Lynch says he won't be directing new episodes of Twin Peaks. Showtime "saddened" over decision, which involved a dispute over money. |
| CTRLsum | (Length bucket 0) **director David Lynch longer direct revival Twin Peaks return 2016 | season | Showtime**: Film director David Lynch will no longer direct the revival of "Twin Peaks" Showtime announced they would produce a third season in October last year. The show was set to return in 2016.

(Length bucket 1) **director David Lynch longer direct revival Twin Peaks set return 2016 | season | Showtime Network saddened**: Film director David Lynch will no longer direct the revival of "Twin Peaks" Showtime Network say they are "saddened" by the decision. The show was set to return for a third season in 2016.

(Length bucket 2) **director David Lynch confirmed longer direct revival Twin Peaks set return 2016 | series | season continue | Showtime Network saddened**: Film director David Lynch has confirmed he will no longer direct the revival of "Twin Peaks" The series was set to return in 2016. Lynch broke the news about his departure in a series of tweets. He said the show's third season will continue without him. Showtime Network said they were "saddened" by Lynch's decision.

(Length bucket 3) **director David Lynch confirmed longer direct revival Twin Peaks 1990s set return 2016 | series | season continue | Showtime Network saddened decision**: SFilm director David Lynch has confirmed he will no longer direct the revival of "Twin Peaks"The 1990s TV series was set to return in 2016. Lynch broke the news about his departure in a series of tweets. The show's third season will continue without him. Showtime Network said they were "saddened" by Lynch's decision.

(Length bucket 4) **director David Lynch confirmed longer direct revival Twin Peaks cult 1990s set return 2016 | series Mark Frost FBI agent murder | tweets season continue | Showtime Network statement saddened decision | Kyle**: Film director David Lynch has confirmed he will no longer direct the revival of "Twin Peaks" The cult 1990s TV series, created by Lynch and Mark Frost, was set to return in 2016. Lynch broke the news about his departure in a series of tweets, saying that the show's third season will continue without him. Showtime Network, which will air the nine-episode comeback, released a statement saying they were "saddened" by Lynch's decision. Actor Kyle MacLachlan, who played the coffee-obsessed FBI agent Dale Cooper in the original series, had confirmed he would reprise the lead role for the new season. |

Table 20: Random Length Control Examples. Control tokens are bolded.

| | |
|---|---|
| Article | Washington (CNN)An Iranian military observation aircraft flew within 50 yards of an armed U.S. Navy helicopter over the Persian Gulf this month, sparking concern that top Iranian commanders might not be in full control of local forces, CNN has learned. The incident, which has not been publicly disclosed, troubled U.S. military officials because the unsafe maneuver could have triggered a serious incident. It also surprised U.S. commanders because in recent months Iranian forces have conducted exercises and operations in the region in a professional manner, one U.S. military official told CNN. "We think this might have been locally ordered," the official said. The incident took place as the U.S. and other world powers meet with Iran in Switzerland to negotiate a deal limiting Tehran's nuclear program. At the same time, Iran has been active in supporting proxies in several hotspots in the Persian Gulf and neighboring regions. The Navy MH-60R armed helicopter was flying from the deck of the USS Carl Vinson on a routine patrol in international airspace, the official said. An unarmed Iranian observation Y-12 aircraft approached. The Iranian aircraft made two passes at the helicopter, coming within 50 yards, before the helicopter moved off, according to the official. The official said the helicopter deliberately broke off and flew away in a 'predictable' manner so the Iranians could not misinterpret any U.S. intentions. The Navy helicopter was in radio contact with the ship during the encounter, but there was no contact between the two aircraft and no shots were fired. The Navy crew took photos of the incident but the military is not releasing them. The U.S. administration is considering a potential demarche protest against Iran, the official said. CNN has reached out to Iranian officials but has not received a response. This type of Iranian observation aircraft generally operates over the Gulf several times a month. But after the recent incident, U.S. naval intelligence did not see it again for two weeks, leading to the conclusion that the incident may have been ordered by a local commander who was then reprimanded by higher-ups. The Pentagon has noted for the last several years that most encounters with the Iranian military at sea or in air are conducted professionally, but that some missions run by Iranian Revolutionary Guard Corps forces have been too aggressive against U.S. forces in the area. The U.S. military's concern has been that one of these incidents could escalate into a military encounter. This incident "might have been buffoonery" the official said, but there is always a risk from such actions. The incident comes as the Navy patrols the Gulf of Aden to watch for Iranian ships the U.S. believes are trying to bring weapons to resupply the Houthi rebels in Yemen. The Navy would share such intelligence with Saudi Arabia, a second U.S. official told CNN. |
| Reference Summary | Iranian plane came within 50 yards of U.S. Navy Sea Hawk copter. Navy copter was on patrol in international airspace. U.S. official think Iranian plane may have been under orders of local commander. |
| CTRLsum | (Length bucket 0) **Iranian military aircraft flew 50 yards armed U.S. Navy helicopter Gulf commanders**: An Iranian military observation aircraft flew within 50 yards of an armed U.S. Navy helicopter over the Persian Gulf this month. The incident, which has not been publicly disclosed, has sparked concern that top Iranian commanders might not be in full control of local forces.

(Length bucket 1) **Iranian military observation aircraft flew 50 yards armed U.S. Navy helicopter Gulf Iranian commanders forces**: An Iranian military observation aircraft flew within 50 yards of an armed U.S. Navy helicopter over the Persian Gulf. The incident sparked concern that top Iranian commanders might not be in full control of local forces.

(Length bucket 2) **Iranian military observation aircraft flew 50 yards armed U.S. Navy helicopter Persian Gulf Iranian commanders control forces | incident**: An Iranian military observation aircraft flew within 50 yards of an armed U.S. Navy helicopter over the Persian Gulf. The incident sparked concern that top Iranian commanders might not be in full control of local forces.

(Length bucket 3) **Iranian military observation aircraft flew 50 yards armed U.S. Navy helicopter Persian Gulf Iranian commanders control forces | incident | Iran | protest**: An Iranian military observation aircraft flew within 50 yards of an armed U.S. Navy helicopter over the Persian Gulf. The incident sparked concern that top Iranian commanders might not be in full control of local forces. Iran has been active in supporting proxies in several hotspots in the region. The U.s. administration is considering a potential demarche protest against Iran.

(Length bucket 4) **Iranian military observation aircraft flew 50 yards armed U.S. Navy helicopter Persian Gulf Iranian commanders control local forces | incident officials | official | ordered | Iran | USS | considering demarche protest | Navy Aden**: An Iranian military observation aircraft flew within 50 yards of an armed U.S. Navy helicopter over the Persian Gulf. The Navy MH-60R armed helicopter was flying from the deck of the USS Carl Vinson on a routine patrol in international airspace. The incident sparked concern that top Iranian commanders might not be in full control of local forces. "We think this might have been locally ordered," the official said. CNN has reached out to Iranian officials but has not received a response. Incident comes as the Navy patrols the Gulf of Aden to watch for Iranian ships. The administration is considering a potential demarche protest against Iran. |

Table 21: Random Length Control Examples. Control tokens are bolded.

| | |
|---|---|
| Article | It's a sight that draws giggles and curious stares from tourists and other first-timers – an unusual festival where revellers carry gigantic phalluses through the streets of a Japanese city. But for the residents of Kawasaki, who lug erotic shapes of all different sizes, this odd tradition is not a joke. Shinto Kanamara Matsuri started as a small tradition but has grown into a popular a tourist attraction, with participants praying to a god of fertility, child birth and protection from sexually transmitted infections. Participants carry a gigantic phallus through the streets of Kawasaki, Japan during the Shinto Kanamara Matsuri festival. The sight of three large phalluses being paraded through neighbourhoods in the city south of Tokyo draws giggles from tourists. Shinto Kanamara Matsuri, the Festival of the Steel Phallus, started as a small tradition but has grown into a popular a tourist attraction. Known as the Festival of the Steel Phallus, it is held every spring at the phallus-shaped Kanayama Shrine. Festivalgoers parade through the streets with three giant phalluses, while spectators lick lollies or snack on sausages or vegetables shaped as male and female genitalia. Rainy weather didn't ruin the mood at this year's festival, which had a massive collection of foreigners, according to Japanese website RocketNews24. They watched as groups of locals carried three heavy phalluses modelled after a mikoshi portable shrine, which is commonly used in Shinto festivals. Residents of Kawasaki carry phalluses of all different sizes while participating in a tradition that began nearly 40 years ago. Participants pray to a god who is said to help with fertility, child birth and protection from sexually transmitted infections. Rainy weather didn't ruin the mood at this year's festival, which attracted thousands of spectators from Japan and abroad. Spectators can buy lollies, sausages and vegetables shaped as male and female genitalia while watching the unusual parade. The three mikoshi carried through the streets for hours are the Kanamara Boat Mikoshi (tall and black), the Elizabeth Mikoshi (big and pink) and the Kanamara Big Mikoshi (squat and wooden), RocketNews24 reported. In addition to the parade there are performances and music, and the festival itself raises money and awareness for HIV research. The festival started nearly 40 years ago, and the Kanayama Shrine was once a popular place for prostitutes to pray for protection from sexually transmitted infections. Today, people visit the shrine to pray for everything from fertility and business prosperity to wedded bliss and good health. In addition to the parade there are performances and music, and the festival itself raises money and awareness for HIV research. The Kanayama Shrine was once a popular place for prostitutes to pray for protection from sexually transmitted infections. Today, people visit the shrine to pray for everything from fertility and business prosperity to wedded bliss and good health. Groups of locals carried three heavy phalluses modelled after a mikoshi portable shrine, which is commonly used in Shinto festivals. |
| Reference Summary | Shinto Kanamara Matsuri started as a small tradition but has grown into a popular tourist attraction. Known as the Festival of the Steel Phallus, participants pray to a god of fertility and protection from infections. Rainy weather didn't ruin the mood at this year's festival, which attracted a large crowd of holidaymakers . |
| CTRLsum | (Length bucket 0) **Shinto Kanamara Matsuri started tradition god fertility child birth protection infections | Steel**: Shinto Kanamara Matsuri, the Festival of the Steel Phallus, started as a small tradition. Participants pray to a god of fertility, child birth and protection from sexually transmitted infections.

(Length bucket 1) **Shinto Kanamara Matsuri started tradition praying god fertility child birth protection sexually infections | Festival Steel**: Shinto Kanamara Matsuri, the Festival of the Steel Phallus, started as a small tradition. Participants pray to a god of fertility, child birth and protection from sexually transmitted infections.

(Length bucket 2) **Shinto Kanamara Matsuri started tradition popular tourist praying god fertility child birth protection sexually infections | Festival Steel Phallus**: Shinto Kanamara Matsuri, the Festival of the Steel Phallus, started as a small tradition but has grown into a popular a tourist attraction. Participants pray to a god of fertility, child birth and protection from sexually transmitted infections.

(Length bucket 3) **Shinto Kanamara Matsuri started tradition popular tourist attraction praying god fertility child birth protection sexually transmitted infections | Festival Steel Phallus**: Shinto Kanamara Matsuri, the Festival of the Steel Phallus, started as a small tradition but has grown into a popular a tourist attraction. Participants pray to a god of fertility, child birth and protection from sexually transmitted infections.

(Length bucket 4) **phalluses | Shinto Kanamara Matsuri started small tradition grown popular tourist attraction praying god fertility child birth protection sexually transmitted infections | Kawasaki Japan Matsuri festival | Tokyo | Festival Steel Phallus | 40**: Shinto Kanamara Matsuri started as a small tradition 40 years ago. It has grown into a popular tourist attraction in Kawasaki, Japan. The Festival of the Steel Phallus is held every spring at the Kanayama Shrine. Participants carry three phalluses through neighbourhoods south of Tokyo. They are praying to a god of fertility, child birth and protection from sexually transmitted infections. Thousands of tourists attended this year's festival, which raised money for HIV research. |

## E.4 CONTRIBUTION SUMMARIZATION ON SCIENTIFIC PAPERS

Here we show three random examples from the arXiv test set. Note that this is the test set from (Cohan et al., 2018) instead of the contribution test data collected by us, because we want to show the difference between reference summaries (i.e. the paper abstract) in existing standard paper summarization dataset and our output contribution summaries. We truncate the source articles since they are too long to display.

Table 22: Random Contribution Summarization Examples. Control tokens are bolded. "[]" denote that the tokens are used as both keywords and prompts.

| | |
|---|---|
| Article | synchronization of neural activity appears in different parts of the mammalian cerebral cortex @xcite , and underlies different neural processes in both normal and anomalous brain functions @xcite . it has been suggested that synchronization plays a vital role in information processing in the brain , e.g. , processing information from different sensory systems to form a coherent and unified perception of the external world @xcite . on the other hand , synchronization has been detected in pathological conditions such as parkinson s disease @xcite . and epileptic seizures have long been considered resulting from excessive synchronized brain activity @xcite , though some recent studies suggest that this picture may be an over - simplification @xcite . therefore understanding the mechanisms of synchronization may be a critical step in elucidating how neural systems work @xcite . it has stimulated a great deal of theoretical and numerical works , such as the studies on the effects of the topological properties of underlying networks @xcite and the dynamical properties of synaptic coupling @xcite . it was recently shown that the response time of synaptic couplings influences the stability of synchronized oscillation in the nonlocally coupled hodgkin - huxley ( hh ) equations @xcite . if the response time of synaptic coupling is slower , synchronized activity of the systems is instable for excitatory coupling . however , the underlying dynamical mechanism of the influence is not clear . in experimental studies @xcite , it has been suggested that the generation of prolonged epileptiform neuronal synchronization is favored by lower efficacy of synaptic transmission . the numerical studies @xcite in a detailed computational model revealed that seizure - like activity occurs when the excitatory synapses are weakened , and the results were confirmed experimentally in mouse neocortical slices . according to the common accepted assumption that synchronization of neuronal activity underlies seizures , the dynamical mechanism of synchronization may be useful for understanding the way the biological neural system works . in this work , we numerically investigated the dynamical mechanism underlying the influence of synaptic efficacy on firing synchronization in hh neuron networks . to do this , we first studied the dynamics of the response of hh neuron to excitatory synaptic current . when the efficacy of synapse is low , namely , strength is weak and duration is short , the limit cycle is stable to the perturbation of the synaptic current . when synaptic efficacy is high , synaptic current can induce the transition of the neurons from limit cycle to fixed point or transient state . the transition is determined by dynamics of neuron s ionic channel . the decrease of synaptic current depresses the feedback of sodium ionic current which is responsible for the initiation of the spike . for simplicity we will refer to the transitions as spike death . in neuronal networks , synaptic input of a neuron is the accumulation of the currents received from all presynaptic neurons . when the coherence of firing time of neurons is enhanced by the excitatory interaction , the synaptic input of neurons transforms from the fluctuating waveform into the pulse shape like the signal produced by one synapse . if synaptic efficacy is high , the input signal can induce spike death of the neuron . then spike death disorders the adjustment of the rhythm of neurons and prevents neurons from firing spikes synchronously . in contrast , for synapses of lower efficacy , the duration of synaptic current is too short to induce spike death of neurons . additionally , the firing synchronization is different from synchronous activity of oscillators for the existence of the transitions of neuron s state . the paper is organized as follows . the hh neuron model and the synaptic coupling were introduced in sec . the response of a hh neuron to synaptic current was investigated in sec . the influence of the dynamics of neurons on firing synchrony was shown in sec . |
| Reference Summary | we investigated the influence of efficacy of synaptic interaction on firing synchronization in excitatory neuronal networks . we found spike death phenomena , namely , the state of neurons transits from limit cycle to fixed point or transient state . the phenomena occur under the perturbation of excitatory synaptic interaction that has a high efficacy . we showed that the decrease of synaptic current results in spike death through depressing the feedback of sodium ionic current . in the networks with spike death property the degree of synchronization is lower and unsensitive to the heterogeneity of neurons . the mechanism of the influence is that the transition of neuron state disrupts the adjustment of the rhythm of neuron oscillation and prevents further increase of firing synchronization . |
| CTRLsum | [**the main contributions of this paper are : ( 1** )]: we investigated the dynamical mechanism underlying the influence of synaptic efficacy on firing synchrony in hodgkin - huxley neuron networks ; ( 2 ) we found that the dynamics of synaptic current plays an important role in determining the stability of firing synchronization . |

Table 23: Random Contribution Summarization Examples. Control tokens are bolded. "[]" denote that the tokens are used as both keywords and prompts.

| | |
|---|---|
| Article | for the understanding of surface reactions and the characterization of materials it is desirable to measure local forces close to a sample surface . the most common method to measure these surface forces is atomic force microscopy ( afm)@xcite . historically , the first force measurements were static measurements for which the force is presented as a scalar function of the static tip - sample separation , the so - called force curve@xcite . this representation is sufficient for conservative forces but the total tip - surface force may also contain contributions from dissipative forces . since dissipative forces depend on probe velocity and past trajectory , dynamic force spectroscopy methods are required for their measurement . moreover , the visualization of dissipative forces as a function of position is valid only for a specific probe trajectory and simple force curves can not capture the full character of the interaction . despite the development of several dynamic methods@xcite surface forces are still usually treated as functions of the probe position only and represented by simple force curves . here , we present a comprehensive framework for the representation and analysis of complex surface forces as they are measured by dynamic afm . we concentrate on the most common modes of dynamic afm : amplitude - modulated afm ( am - afm ) and frequency - modulated afm ( fm - afm ) , which can be considered as narrow frequency band methods@xcite . we explore the fundamental limit of force reconstruction with narrow band dynamic afm at fixed probe height and show how minimal assumptions allow for a quantitative reconstruction of the tip - surface interaction . at the heart of the afm apparatus is a micro - cantilever with a sharp tip . the cantilever is firmly clamped at one end and the tip is located at the other end which can move freely . it is assumed that surface forces only act on the tip whereas the rest of the cantilever does not experience significant surface forces . in dynamic afm an additional external drive force is applied to maintain an oscillatory motion . thus , the dynamics are governed by the force between tip and surface , the external drive force and the properties of the cantilever beam . since the cantilever is a three dimensional continuum object its motion is usually described by the amplitudes of different oscillation eigenmodes . in general , these modes can cause the cantilever to bend in all directions in space . however , the cantilever is positioned such that the softest flexural modes bend the beam in a plane orthogonal to the surface plane . we restrict ourselves to the case where only these flexural modes are excited by the drive force . due to this experimental configuration the cantilever is much more susceptible to the component of the tip - surface force which is orthogonal to the surface plane . this component of the force is typically the most dominant component and the influence of lateral force components is considered negligible . in this case the cantilever acts as a mechanical projector which reacts only to one component of a three dimensional force vector field . the deflection @xmath0 of a cantilever of length @xmath1 orthogonal to surface is described by a one dimensional euler - bernoulli equation@xcite @xmath2 where @xmath3 is the young s modulus , @xmath4 is the second moment of area , @xmath5 is the mass per unit length of the cantilever , @xmath6 is the position coordinate along the cantilever beam and @xmath7 is the time variable . the force term @xmath8 includes the surface forces acting as a point - like load at position @xmath9 , the external drive force and the hydrodynamic damping due to the surrounding medium@xcite . |
| Reference Summary | in atomic force microscopy ( afm ) tip - surface interactions are usually considered as functions of the tip position only , so - called force curves . however , tip - surface interactions often depend on the tip velocity and the past tip trajectory . here , we introduce a compact and general description of these interactions appropriate to dynamic afm where the measurement of force is restricted to a narrow frequency band . we represent the tip - surface interaction in terms of a force disk in the phase space of position and velocity . determination of the amplitude dependence of tip - surface forces at a fixed static probe height allows for a comprehensive treatment of conservative and dissipative interactions . we illuminate the fundamental limitations of force reconstruction with narrow band dynamic afm and we show how the amplitude dependence of the fourier component of the force at the tip oscillation frequency , gives qualitative insight into the detailed nature of the tip - surface interaction . with minimal assumptions this amplitude dependence force spectroscopy allows for a quantitative reconstruction of the effective conservative tip - surface force as well as a position - dependent damping factor . we demonstrate this reconstruction on simulated intermodulation afm data . _ keywords _ : atomic force microscopy , measurement of force , mechanical resonators , mems / nems , dissipation , intermodulation |
| CTRLsum | [**the main contributions of this paper are : ( 1 )**]: a comprehensive framework for the representation and analysis of complex surface forces as they are measured by dynamic atomic force microscopy ( afm ) ; ( 2 ) a study of the fundamental limit of force reconstruction with narrow band dynamic afm at fixed probe height and show how minimal assumptions allow for a quantitative reconstruction of the tip - surface interaction . |

Table 24: Random Contribution Summarization Examples. Control tokens are bolded. "[]" denote that the tokens are used as both keywords and prompts.

| | |
|---|---|
| Article | in this paper we discuss the mathematical aspects of the problems originating in the solution of nonlinear systems of hyperbolic partial differential equations . these equations describe a large variety of physical phenomena , such as , gasdynamics , magnetohydrodynamics ( mhd ) , shallow water equations , elasticity equations , etc . being nonlinear , these systems usually require numerical methods for their solution . presence of discontinuous solutions motivates the necessity of the development of reliable numerical methods based on the fundamental mathematical properties of hyperbolic systems . although such methods are rather well developed for the euler gasdynamic equations in the conservation law form , their extension to more complicated hyperbolic systems is not straightforward . it requires a mathematical justification of the solution uniqueness , a formulation of the selection principles for relevant solutions , and , finally , an investigation of their physical validity . most of high - resolution methods for gasdynamic equations use the exact or some of the approximate self - similar riemann problem solutions to determine fluxes through the computational cell surfaces . similar methods are expected to be developed for various types of hyperbolic systems . in this case we must construct the elementary self - similar solution using only admissible discontinuities ( entropy consistent , evolutionary , etc . ) . basically the choice of the solution must be made on the basis of the structure of the solution of the extended problem @xcite . all mentioned above makes very important the study of discontinuous solutions behavior under vanishing viscosity and dispersion to create a proper background for the development of high - resolution numerical methods for hyperbolic systems more complicated than the euler equations of gasdynamics . we discuss several analytical and numerical solutions in the mentioned fields which illustrate the complexity of the selection problem and outline the methods of its solution . tvd upwind and symmetric differencing schemes have recently become very efficient tool for solving complex multi - shocked gasdynamic flows . this is due to their robustness for strong shock wave calculations . the extension of these schemes to the equations of the ideal magnetohydrodynamics is not simple . first , the exact solution @xcite of the mhd riemann problem is too multivariant to be used in regular calculations . second , several different approximate solvers @xcite , @xcite , @xcite , @xcite , @xcite , @xcite , and @xcite applied to mhd equations are now at the stage of investigation and comparison . this investigation requires i ) determination of a proper slope limiting method in the parameter interpolation procedure necessary to obtain nonoscillatory schemes of the order of accuracy higher than one ; ii ) development of an efficient entropy correction method necessary to exclude rarefaction shocks ; and , finally , iii ) solution of the problem of excluding the origin of nonevolutionary solutions in ideal mhd calculations . the system of governing equations for a mhd flow of an ideal , infinitely conducting , perfect plasma in the cartesian coordinate system @xmath0 , @xmath1 , @xmath2 with the use of the conventional notations reads ( one fluid approximation ) : @xmath3 where @xmath4 is the vector of conservative variables and @xmath5 , @xmath6 , and @xmath7 are the flux vectors . we introduced here the source term @xmath8 in the form @xmath9 this form of the system can be used to satisfy the divergence - free condition by convecting away the magnetic charge from the computational region @xcite . otherwise , any other well - known method can be used to eliminate the magnetic charge . to determine a numerical flux @xmath10 normal to the computational cell boundary ( @xmath11 is a unit outward vector normal to the cell surface ) one can use the formulas based on the solution of the linearized problem @xmath12 . ] ] here @xmath13 and @xmath14 are the matrices formed by the right and by the left eigenvectors , respectively , of the frozen jacobian matrix @xmath15 the matrix @xmath16 is a diagonal matrix consisting of the frozen jacobian matrix eigenvalue moduli . the superscripts @xmath17 and @xmath18 denote the values at the right- and at the left - hand side of the cell boundary . |
| Reference Summary | a number of physical phenomena are described by nonlinear hyperbolic equations . presence of discontinuous solutions motivates the necessity of development of reliable numerical methods based on the fundamental mathematical properties of hyperbolic systems . construction of such methods for systems more complicated than the euler gas dynamic equations requires the investigation of existence and uniqueness of the self - similar solutions to be used in the development of discontinuity - capturing high - resolution numerical methods . this frequently necessitates the study of the behavior of discontinuities under vanishing viscosity and dispersion . we discuss these problems in the application to the magnetohydrodynamic equations , nonlinear waves in elastic media , and electromagnetic wave propagation in magnetics . |
| CTRLsum | [**the main contributions of this paper are : ( 1 )**]: the mathematical aspects of the problems originating in the solution of nonlinear systems of hyperbolic partial differential equations ; ( 2 ) the study of discontinuous solutions behavior under vanishing viscosity and dispersion to create a proper background for the development of high - resolution numerical methods for hyperbola systems more complicated than the euler equations of gasdynamics ; and ( 3 ) solution of the problem of excluding the origin of nonevolutionary solutions in ideal magnetohydrodynamics calculations . |

## E.5 Invention Purpose Summarization on Patent Filings

Here we show three random examples from the BIGPATENT test set. Note that this is the test set from origial BIGPATENT, because we want to show the difference between reference summaries in existing standard dataset and our output purpose summaries. We truncate the source articles since they are too long to display.

Table 25: Random Invention Purpose Summarization Examples. Control tokens are bolded. "[]" denote that the tokens are used as both keywords and prompts.

| | |
|---|---|
| Article | referring to the drawings and , in particular to fig1 therein illustrated is a prior art surgical support mesh 10 . mesh 10 may be manufactured from monofilament or multifilament yarns . prior art mesh 10 , as shown , includes multifilament horizontally - extending yarns 12 and multifilament vertically - extending yarns 14 woven together to form a support trellis . the use of multifilament yarns , such as yarns 12 and 14 , provides a mesh having greater pliability and suppleness than the use of monofilament yarns . these characteristics result from both the smaller diameter of the individual filaments and the interstitial spaces or voids that are located between such filaments . in particular , the flexibility of a filament ( or fiber ) generally increases as its diameter decreases . because the solid cross - sectional area of the filaments of a multifilament yarn is less than the cross - sectional area of a monofilament yarn of equivalent diameter , the multifilament yarn will have a greater degree of flexibility and pliability than that of the monofilament yarn . as shown in fig1 a , each of multifilament yarns 12 and 14 is composed of a plurality of filaments 16 that are intermingled or bundled together to form the yarn . interstitial spaces 18 , which are pockets of air , are formed between adjacent filaments of the yarn . although these voids contribute to the softness and pliability of the formed mesh , they also provide a natural breeding ground for bacteria or other infectious material . surgical mesh is , of course , thoroughly sterilized prior to implantation . nevertheless , surgeons typically prefer the use of monofilament - designed mesh to minimize any risk of infection . as a result , the advantages associated with multifilament - designed mesh ( i . e ., softness and pliability which result in better assimilation of the mesh into the body ) are typically sacrificed . it has been discovered herein that a surgical support mesh having both the softness and pliability of a multifilament - designed mesh and the infection resistance of a monofilament - designed mesh may be produced . particularly , it has been discovered that a support trellis formed of multifilament yarn wherein the interstitial voids located between adjacent filaments are enclosed within an infection - impervious matrix exhibits the desired resistance to harboring of infectious matter without significant loss of flexibility . particularly , the matrix , which completely encloses the interstitial voids between the filaments of the yarn , provides an effective barrier to the passage of infectious matter between the interior and exterior of the yarn . accordingly , any voids remaining in the yarn after encapsulation of such yarn are enclosed ( and thereby sealed ) within the resultant matrix . a first embodiment of the present invention is shown in fig2 . particularly , this first embodiment includes a support trellis 20 formed of multifilament yarns 22 and 24 which overlap at cross - over junctions 25 . subsequent to forming of the trellis , such trellis is encapsulated within a matrix 26 , which is preferably a flexible material that continuously surrounds the exterior of the yarns thereby enclosing interstitial voids 27 located between filaments 28 ( see fig2 a ). in one embodiment , the matrix is formed from a polymeric resin . as shown in fig2 a , the resin can be applied to the yarn in such a manner as to not allow the resin to substantially penetrate into the yarn . particularly , the penetration of the resin can be controlled through the application procedure , e . g ., quantity of resin applied and / or encapsulating time . in such an embodiment , the interstitial spaces are enclosed ( rather than filled ) within the continuous matrix . however , it is contemplated that the resin can be allowed to penetrate into the yarn , thereby substantially filling the void space located therein . in another embodiment of the present invention , individual yarns 29 , as shown in fig3 are encapsulated within matrix 30 prior to forming of the support trellis . fig3 a shows a compressed yarn 29 which provides a trellis having a reduced thickness . as a result of the encapsulation , interstitial voids 32 remaining in the yarn are enclosed ( and thereby sealed ) within the matrix . |
| Reference Summary | a soft and pliable surgical support mesh exhibiting increased resistance to inhabitation of infectious matter . the mesh includes a support trellis formed of multifilament yarns wherein the interstitial voids located between the filaments of said yarns are enclosed within an infection - impervious matrix . the meshes may be designed to be extremely thin yet retain the requisite strength for repairing soft tissue , which allows for a low profile when folded for delivery . |
| CTRLsum | [**the purpose of the present invention is**]: to provide a surgical mesh that is resistant to the growth of bacteria and other infectious matter . this is accomplished by encapsulating the interstitial spaces located between the filaments of the yarn within a matrix . |

Table 26: Random Invention Purpose Summarization Examples. Control tokens are bolded. "[]" denote that the tokens are used as both keywords and prompts.

| | |
|---|---|
| Article | fig1 shows a multicolor web fed rotary printing press 1 in accordance with the invention . the press 1 includes four tower arrangements 2 a , 2 b , 2 c and 2 d for printing a single color or a multicolor image on the webs 4 a , 4 b , 4 c and 4 d . the webs 4 a , 4 b , 4 c and 4 d travel in a substantially linear direction through each of the towers 2 a - 2 d . for example , the web can travel along a substantially vertical path , as shown in fig1 . alternatively , as those skilled in the art will appreciate , the web path can be in a substantially horizontal direction , or in a substantially linear path at any desired angle relative to the vertical direction shown . the towers 2 a - 2 d each include four printing units 6 c , 6 m , 6 y and 6 b for respectively printing an image in cyan , magenta , yellow and black on both sides of each web 4 a - 4 d . other colors besides cyan , magenta , yellow and black can be used . the webs 4 can be , for example , between 1200 and 1600 millimeters wide . each of the printing units 6 c , 6 m , 6 y and 6 b in a tower can be moved along a respective web 4 by a lifting and positioning system 8 shown in fig2 . the lifting and positioning system 8 includes a spindle drive 10 , which has a fixed spindle 12 spanning a range 14 over which the printing units 6 c , 6 m , 6 y and 6 b can be moved . each of the printing units 6 c , 6 m , 6 y and 6 b includes a ball screw 16 , which is rotatably supported in a housing 18 . the ball screw 16 can be rotated by a motor 20 as shown in fig2 . fig2 shows one set of a spindle drive 10 , fixed spindle 12 , ball screws 16 , and motors 20 , but preferably each tower 2 is provided with several sets , one set for each corner of the print unit housing 18 . the motors 20 are controlled by a motor control unit 22 , which receives commands from a remote control 24 . by pressing a button on the remote control 24 , an operator can control the rotation of the motors 20 and thereby the movement direction and position of each printing unit 6 b , 6 y , 6 m and 6 c in a tower 2 . rail systems ( not shown ) fixed to a side frame of each tower 2 can also be used to precisely guide movements of the printing units 6 in the tower . as shown in fig1 and 2 , the position of each of the printing units 6 along the webs 4 and fixed spindles 12 can be controlled by the operator to allow access to a desired part of a printing unit 6 . for example , in fig1 the operator has moved the print unit 6 b of tower 2 b into a position where a printing plate of the print unit 6 b can be most easily accessed . after the printing plate has been accessed , the operator can move the print unit 6 b into a different position so that inker units in an upper part of the printing unit 6 b can be easily accessed . two or more printing units 6 in tower 2 can also be moved as a group . for example , if the operator wants to access the plate cylinder of the printing unit 6 m of tower 2 b shown in fig1 he can simply move the two printing units 6 y , 6 m upwardly together until the top surface 28 of the printing unit 6 y contacts the bottom surface 30 of the printing unit 6 b . thereafter , the operator can move the group of printing units 6 b , 6 y and 6 m upwards into the position shown in tower 2 c of fig1 where the plate cylinder of the printing unit 6 m can be easily accessed . although fig1 and 2 show four printing units 6 for each tower 2 , different numbers of printing units can be used for each tower . fig1 shows that the lower three printing units 6 of the tower 2 b are beneath an operating floor 26 . preferably at least two of the printing units can be lowered beneath the operating floor 26 , and a printing press in accordance with the invention can be configured so that all of the printing units in a tower can be lowered beneath an operating floor . the operating floor can be a floor of a print shop , or can be an elevated platform . |
| Reference Summary | the present invention is directed to a multicolor web fed rotary printing press having printing units that can be moved along a linear section of a web by a positioning mechanism to allow easy access to each of the printing units . the easy access to the printing units significantly reduces maintenance costs and downtime of the press . in addition , the invention provides a printing press that has a reduced overall size and allows the printing units to be arranged in a nested formation during printing operations . this nested formation reduces fanout and paper waste . the linear web section can be oriented vertically or horizontally . |
| CTRLsum | [**the purpose of the present invention is**]: to provide a web fed rotary printing press that allows an operator to access a desired part of a printing unit more easily than in the past . |

Table 27: Random Invention Purpose Summarization Examples. Control tokens are bolded. "[]" denote that the tokens are used as both keywords and prompts.

| | |
|---|---|
| Article | referring now to the drawings and more particularly to fig1 - 3 , a fluid system 10 is illustrated and includes a variable displacement axial piston pump 12 that receives fluid from a tank 14 via a conduit 16 and delivers pressurized fluid via a supply conduit 18 to a fluid control valve 20 and selectively through work conduits 22 , 24 to a fluid actuator 26 . in the subject arrangement , the variable displacement axial piston pump 12 is a unidirectional pump that rotates in a counterclockwise direction as driven by a power input shaft 27 . the fluid system 10 also includes first and second pressure sensors 28 , 30 respectively connected to the tank conduit 16 and the supply conduit 18 . the pressure sensors 28 , 30 are operative to sense the pressure in the respective lines and deliver an electrical signal to a controller 32 through electrical lines 34 , 36 . a position sensor 40 is mounted on the variable displacement axial piston pump 12 and operative to sense the displacement of the pump and deliver a signal representative thereof to the controller 32 via an electrical line 42 . various other components could be used in the subject fluid system 10 without departing from the essence of the subject invention . for example , several control valves 20 and associated fluid actuators 26 could be used . likewise , other sensors of various types and styles could be used . the variable displacement axial piston pump 12 includes a housing 44 having a head portion 46 and a body portion 48 . the head portion 46 defines an inlet port passage 50 that is connected to the conduit 16 and an outlet port passage 52 that is connected to the supply conduit 18 . in the subject arrangement , a port plate 54 is disposed between the head portion 46 and the body portion 48 . the construction of the porting within the port plate 54 is more clearly illustrated in fig3 and will be discussed more fully below . it is recognized that the porting illustrated in fig3 could be made within the head portion 46 without departing from the essence of the subject invention . a rotating group 56 is disposed within the body portion 48 and includes a barrel 58 having a plurality of cylinder bores 59 defined therein spaced from one another around an axis of rotation 60 of the barrel 58 . each of the cylinder bores 59 is oriented within the barrel 58 parallel with the axis of rotation 60 . a plurality of piston assemblies 62 are operatively associated with the barrel 58 and each one of the plurality of piston assemblies 62 includes a piston 64 slideably disposed in the respective ones of the plurality of cylinder bores 59 . each one of the plurality of piston assemblies 62 also has a shoe 66 pivotally attached to one end of each piston 64 in a conventional manner . the barrel 58 has an end surface 68 that is in mating , sealing contact with the port plate 54 to provide communication between the cylinder bores 58 and the respective inlet and outlet port passages 50 , 52 of the head portion 46 . a closed chamber 70 is defined in each cylinder bore 59 of the barrel 58 between the end of the piston 64 and the end surface 68 thereof . referring to fig3 the porting between the barrel 58 and inlet and outlet port passages 50 , 52 of the head portion 46 is more clearly illustrated . for explanation purposes only , the " 270 " degree position illustrated in fig3 relates to a position on the right side of the drawing of fig1 and the " 0 " degree position illustrated in fig3 relates to a position on the right side of the drawing of fig2 . an arcuate slot 72 is defined in the port plate 54 and provides communication between the plurality of closed chambers 70 and the inlet port passage 50 . a plurality of slots 74 are defined in the port plate 54 circumferentially spaced from the arcuate slot 72 and provides communication between the plurality of closed chambers 70 and the outlet port passage 52 . |
| Reference Summary | a variable displacement axial piston pump is typically used to receive fluid from a tank and supply pressurized fluid through a control valve to move an actuator . the present variable displacement axial piston pump has a swashplate arrangement that is capable of being angled in two different directions to control the pressure transitions between the low pressure inlet port passage and the higher pressure outlet port passage as cylinder bores in a barrel of a rotating group rotate through trapped volume regions situated between inlet and outlet port passages of the axial piston pump . movement of the swashplate arrangement in two different directions provides smooth pressure transitions and increases the operating efficiency of the variable displacement axial piston pump . |
| CTRLsum | [**the purpose of the present invention is**]: to provide a variable displacement axial piston pump that is capable of delivering a variable amount of pressurized fluid in response to a change in the displacement of the pump . |

## E.6 QUESTION-GUIDED SUMMARIZATION

We randomly sample 3 articles from NewsQA and show five questions and answers from CTRLsum for each article. We also show the gold answers to these questions.

Table 28: Random Examples on Question-guided summarization. Control tokens are bolded. "[]" denote that the tokens are used as both keywords and prompts.

| | |
|---|---|
| Article | TEHRAN, Iran (CNN) – Iran's parliament speaker has criticized U.S. President-elect Barack Obama for saying that Iran's development of a nuclear weapon is unacceptable. Iranian President Mahmoud Ahmadinejad has outlined where he thinks U.S. policy needs to change. Ali Larijani said Saturday that Obama should apply his campaign message of change to U.S. dealings with Iran. "Obama must know that the change that he talks about is not simply a superficial changing of colors or tactics," Larijani said in comments carried by the semi-official Mehr News Agency. "What is expected is a change in strategy, not the repetition of objections to Iran's nuclear program, which will be taking a step in the wrong direction." In his first post-election news conference Friday afternoon, Obama reiterated that he believes a nuclear-armed Iran would be "unacceptable." He also said he would help mount an international effort to prevent it from happening. Larijani said that U.S. behavior toward Iran "will not change so simply" but that Obama's election showed internal conditions in the United States have shifted. He added that Iran does not mind if the United States provides other Persian Gulf countries with nuclear technology, but "you should know that you cannot prevent the Islamic Republic [from reaching its goals in the nuclear field]," according to the news agency. Obama cautioned Friday that it had only been a few days since the election and that he was not in office. "Obviously, how we approach and deal with a country like Iran is not something that we should simply do in a knee-jerk fashion. I think we've got to think it through," Obama said. "But I have to reiterate once again that we only have one president at a time. And I want to be very careful that we are sending the right signals to the world as a whole that I am not the president, and I won't be until January 20th." Larijani was speaking two days after Iranian President Mahmoud Ahmadinejad congratulated Obama, the first time an Iranian leader has offered such wishes to a U.S. president-elect since the 1979 Islamic Revolution. One analyst said the welcome was a gesture from the hard-line president that he is open to a more conciliatory relationship with the United States. Ahmadinejad said Tehran "welcomes basic and fair changes in U.S. policies and conducts," according to the state-run Islamic Republic News Agency on Thursday. Relations between the United States and Iran have historically been chilly and have been further strained in recent years over Iran's nuclear program. Tehran insists that the program exists for peaceful purposes, but the United States and other Western nations are concerned by Iran's refusal to halt uranium enrichment activities. CNN's Shirzad Bozorgmehr contributed to this report. |
| CTRLsum | [**Q: Iran criticizes who? A:**]: U.S. President-elect Barack Obama.
*Gold Answer: U.S. President-elect Barack Obama*

[**Q: What are US and Iran relations tensioned about? A:**]: Iran's nuclear program.
*Gold Answer: nuclear program.*

[**Q: Who said Obama should apply campaign message? A:**]: Ali Larijani, Iran's parliament speaker.
*Gold Answer: Ali Larijani*

[**Q: What should Obama apply according to speaker? A:**]: Change in strategy, not the repetition of objections.
*Gold Answer: his campaign message of change*

[**Q: What kind of weapons are being discussed? A:**]: Nuclear weapons.
*Gold Answer: nuclear* |

Table 29: Random Examples on Question-guided summarization. Control tokens are bolded. "[]" denote that the tokens are used as both keywords and prompts.

| Article | Editor's note: The staff at CNN.com has recently been intrigued by the journalism of VICE, an independent media company and website based in Brooklyn, New York. VBS.TV is Vice's broadband television network. The reports, which are produced solely by VICE, reflect a transparent approach to journalism, where viewers are taken along on every step of the reporting process. We believe this unique reporting approach is worthy of sharing with our CNN.com readers. Brooklyn, New York (VBS.TV) – Whenever someone at our office bitches about being overworked, our stock response is "Beats digging ditches." While the express intention of the statement is usually not-so-supportive, we think it's a healthy reminder that at the end of the day, we are all basically professional e-mailers and should be thankful for such. The wildland firefighters who work for Grayback Forestry in Medford, Oregon, have no such motivational adages because their job is actually digging ditches. Around active forest fires. On the sides of mountains. You can't even bitch at these guys for having cushy government pensions to fall back on when they get older, because they're all private-sector contractors. Which means if they aren't out fighting forest fires or doing preventative forestry on unburned woods (basically extreme landscaping), they are losing money. They are the hardest working men in the tree business. Southern Oregon in the summer is a tinderbox. Last year the state recorded some 560 wild fires, the majority of which occurred in the seemingly endless sea of trees running across its bottom from the Cascades to the Pacific Coast. Humidity is next to nonexistent, which is extremely pleasant, but means that even an errant spark from a chainsaw or the proverbial cigarette butt out the car window can set the entire region ablaze. Flying into the Rogue Valley, there is evidence of past wild fires is everywhere: From the miles-long scar of the 2002 "Biscuit" fire stretching past the horizon, to the smaller pockets of charred trees crowding the edge of towns to the blue Wilderness-Firefighter-ribbon bumper stickers flying past on the I-5 to the elaborate wildfire and firefighter shrines at a local bar. See more of Oregon Fire Lines at VBS.TV There are very few places firemen aren't revered as local heroes, but the wilderness firefighters of Oregon go past people simply risking their lives to help others into a crazy superhero realm where their work regularly prevents entire settlements from being destroyed. They're literally the guardians of their communities. Incidentally, their work is also very beneficial for the forest. We spent a few days following a crew of Grayback forest-firefighters walk up the sides of what most people would consider a cliff, to chop down underbrush in preparation for a controlled burn. This is what they like to call "project work" – the light stuff they do between fires. The work is the hardest and least rewarding work we have ever tried to do. Unless you consider 12-hour-plus shifts of backbreaking labor, virtually zero outside recognition, and occasional accusations of being shills for the timber industry rewards. Which we do not. This piece was originally produced in August 2010. |
|---|---|
| CTRLsum | [ **Q: What does the VBS call it? A:**]: Project work. 
 *Gold Answer: "Beats digging ditches."* 

 [**Q: How long are the shifts? A:**]: 12-hour shifts. 
 *Gold Answer: 12-hour-plus* 

 [**Q: What does the 12 hr shift involved? A:**]: Project work. 
 *Gold Answer: backbreaking labor, virtually zero outside recognition, and occasional accusations of being shills for the timber industry* 

 [**Q: Which group followed firefighters? A:**]: The wildland firefighters of Oregon. 
 *Gold Answer: VBS.TV* 

 [**Q: What is the VBS following? A:**]: Wildland firefighters who work around active forest fires. 
 *Gold Answer: a crew of Grayback forest-firefighters* |

Table 30: Random Examples on Question-guided summarization. Control tokens are bolded. "[]" denote that the tokens are used as both keywords and prompts.

| | |
|---|---|
| Article | WASHINGTON (CNN) – The nation's largest publicly owned utility company may be vulnerable to cyber attacks, according to a new report. In 2007 President Bush visited the Browns Ferry Nuclear Plant, operated by the Tennessee Valley Authority. The Tennessee Valley Authority, which supplies power to almost 9 million Americans, "has not fully implemented appropriate security practices to protect the control systems used to operate its critical infrastructures," leaving them "vulnerable to disruption," the Government Accountability Office found. Simply put, that means a skilled hacker could disrupt the system and cause a blackout. Rep. James Langevin, a Rhode Island Democrat, fears the problem is much larger than just the TVA. "If they are not secure, I don't have a great deal of confidence that the rest of our critical infrastructure on the electric grid is secure," he said. The TVA operates 52 nuclear, hydroelectric and fossil-fuel facilities in the southeastern United States. Among the government watchdog agency findings: ● The TVA's firewalls have been bypassed or are inadequately configured ● Passwords are not effective ● Servers and work stations lack key patches and effective virus protection ● Intrusion-detection systems are not adequate ● Some locations lack enough physical security around control systems. The GAO recommends 73 steps to correct the problems in its report to Congress. In September, CNN first aired dramatic footage of a government experiment demonstrating that a cyber attack could destroy electrical equipment. The experiment, dubbed "Aurora," caused a generator to fall apart and grind to a halt after a computer attack on its control system. The test was conducted by scientists at the Idaho National Laboratory. In October, the North American Electric Reliability Corp. told Congress that 75 percent of utilities had taken steps to mitigate the Aurora vulnerability, but Langevin said it now appears that Congress was misled. A congressional audit of the electric reliability corporation's claim cast doubt on the assertion that most utilities were taking steps to fix the problem. "It appears that they just made those numbers up," Langevin said. "It is not acceptable. It is outrageous." He said the result is there is now no clear picture of how vulnerable utilities are to cyber attacks. The electric reliability corporation – a nongovernmental group that oversees the power system and comprises members of the industry and some consumers – told CNN it regrets the confusion. Experts told CNN that Cooper Industries is the only manufacturer of hardware that can close the Aurora vulnerability. The company estimated it would need to sell about 10,000 devices to fix the problem nationwide. It has sold just over 100, it told CNN. Langevin said the federal government may need new powers to require utilities to take corrective actions to close cyber security gaps, and he will press to give those powers to the Federal Energy Regulatory Commission. The congressman is chairing an Emerging Threats, Cybersecurity, and Science and Technology subcommittee hearing Wednesday afternoon. Representatives of the TVA, the GAO, the federal commission and the electric reliability corporation are to appear before the subcommittee. |
| CTRLsum | [**Q: Are cyber attacks a real threat? A:**]: Yes. *Gold Answer: skilled hacker could disrupt the system and cause a blackout.*

[**Q: What steps are being taken to access vulnerabilities? A:**]: The GAO recommends 73 steps. *Gold Answer: 73*

**Q: What percentage of utilities were fixed to combat attacks? A:**]: 75 percent. *Gold Answer: 75 percent*

[**Q: Who can disrupt the Tennessee Valley Authority? A:**]: A skilled hacker. *Gold Answer: skilled hacker*

[**Q: What was Congress told? A:**]: Nothing. *Gold Answer: 75 percent of utilities had taken steps to mitigate the Aurora vulnerability,* |

