# OpenReview forum: "CTRLsum: Towards Generic Controllable Text Summarization"
_ICLR.cc/2021/Conference — Reject_

### Official Review · AnonReviewer1 · 2020-10-28
**Official Blind Review #1**

**Rating:** 6
**Confidence:** 4

**Review:**

Paper Summary:
* This paper proposes a framework for controllable summarization, CTRLsum.  It is different from standard summarization models that CTRLsum uses a set of keywords extracted from the source text automatically or descriptive prompts to control the summary.  Experiments with three domains of summarization datasets and five control aspects.

Strengthes:
* The authors investigated the effectiveness of the proposed model through extensive experiments.

Weaknesses:
* The proposed method that uses keywords as an additional input text is almost the same as CIT (Saito et al., 2020), and the scores of CTRLsum on the CNNDM dataset reported in Table 7 does not outperformed those of CIT.  Also, it is not novel to use descriptive prompts to control natural language generation.
   * Saito et al.: Abstractive Summarization with Combination of Pre-trained Sequence-to-Sequence and Saliency Models. CoRR abs/2003.13028 (2020)
* I think that the author's claim, "keywords and prompts are complementary", is not evaluated fully.

Questions:

* With respect to contribution summarization, did you evaluate CTRLsum(keyword without prompt) and CTRLsum(prompt without keywords)?  The control tokens "the main contributions of this paper are : ( 1 )" is far from the keywords used during training, and so I think that the keywords are not effective for contribution summarization.  In fact, BART that uses prompt worked well for contribution summarization.

* Did you evaluate the ablation tests with respect to the special token "|" and keyword dropout?

* Can CTRLsum control the generation with multiple aspects (length and entity control, length and QA control, etc.) simultaneously?  The length of  summaries generated by CTRLsum is strongly dependent the number of keywords, and so I think it is difficult to simultaneously control multiple aspects including length control.

Update:
Thank you for the answers to my questions and additional experiments.

---

> ### Author Response · Authors · 2020-11-24
> **We have added ablation results on the role of keywords and prompts for CTRLsum**
>
> We thank the reviewer for the time and comments. Due to time limitations, we could only address major points, but we’ll try to reflect all advice in future revisions.
>
>
> > Q1: The proposed method that uses keywords as an additional input text is almost the same as CIT (Saito et al., 2020)
>
> Thank you for the pointer, we were not aware of this relatively recent paper. We have added the paper to the Related Works section in the updated version, however, the paper is still "work in progress" as noted by the original authors and was not published at a peer-reviewed venue, thus we will refrain from comparing our results with the mentioned work.
>
> We argue that the main point of our paper is for controllable summarization and the strong performance on uncontrolled summarization is a side product of CTRLsum. The formulation of a framework that controls summarization through control tokens and quantitatively evaluating the effectiveness in five control dimensions stands as one of the main contributions of this paper.
>
> > Q2: the author's claim, "keywords and prompts are complementary", is not evaluated fully.
>
> We have added new ablation results covering CTRLsum (keyword only), CTRLsum (prompt only), and CTRLsum (keyword + prompt) into Appendix C to provide stronger support to the claim, “keywords and prompts are complementary”. Please check the paper for details. We summarize several evidence that backs this claim: (1) in the previous results, the BART baseline with prompts performs poorly for entity and length control (e.g. with full article entity success rate < 20%), comparably well to CTRLsum (keyword + prompt) for contribution summarization, greatly worse than CTRLsum (keyword + prompt) on purpose summarization and QA. (2) In the newly added results on entity control, contribution, and QA, using keywords alone is critical for entity control success, using prompts (either alone or with keywords) is important for contribution summarization, using prompts and keywords together is crucial for QA.
>
> In conclusion, although for a specific control task one of keywords and prompts might dominate over another, neither keywords nor prompts could succeed alone in all dimensions. Therefore, we think they are complementary for the sake of general and flexible controllable summarization. Hope these new results address your concerns!
>
> > Q3: With respect to contribution summarization, did you evaluate CTRLsum(keyword without prompt) and CTRLsum(prompt without keywords)?
>
> We added this result in Appendix C, Table 11. You are correct that keywords are not effective for contribution summarization.
>
> > Q4: Did you evaluate the ablation tests with respect to the special token "|" and keyword dropout?
>
> We had some experience with these two model variants in the early stage of this project but could not find these two model checkpoints, thus we cannot provide fully ablation results at this time due to time limitation. However, we did have some impressions about them which hopefully answer your questions a bit: (1) the special token “|” does not matter much and it does not show an obvious correlation with the summary. (2) keyword dropout training does not have evident effect on uncontrolled summarization scores with automatic keywords, but without keyword dropout we observed that the summary demonstrates a stronger correlation with the keywords and weaker dependence on the article. We were worried that this may hurt control since it is often desired that the summarization system can retrieve related information from the article related to the keywords, yet we didn’t do further quantitative evaluation on this.
>
> > Q5: Can CTRLsum control the generation with multiple aspects
>
> This is an interesting point. In our method, some control aspects can be difficult to be combined with others, for example, the QA and others are hard to combine given that we are using a reading-comprehension-style prompt. Combining entity and length is possible under the CTRLsum model, but that requires a more sophisticated control logic between users and control tokens. For instance, the current control logic of entity control is simply using that entity words as the control tokens, the control logic of length control is to vary the number of automatic keywords as control tokens given user length preference. Such control logic can be more complicated -- for example, identifying the related words for a given entity (such as the words which appear in the same sentences of the entity for simplicity) and selecting K of these words with the highest selection probability by the tagger (K depends on user length input) can probably do the job of combining entity and length. The underlying CTRLsum model keeps unchanged regardless of the test-time control logic. We think such directions are definitely worth exploring in the future for more flexible controllable summarization.

---

### Official Review · AnonReviewer2 · 2020-10-28
**A simple but effective method of focusing abstractive summarization models.**

**Rating:** 7
**Confidence:** 4

**Review:**

The authors propose an abstractive document summarization model that can
generate summaries that target a specific set of keywords or prompts. This is
in contrast to generic summarization models that learn to summarize a document
but are difficult to control or direct. The authors propose a straightforward
way of obtaining keywords from an article similar in spirit to Gerhmann et al.
2018. Alternatively, "ground truth" keywords can be found using a reference
summary. In either case, the keywords are prepended to the input document and
a BART model is fine-tuned to generate summaries using both the document and
keyword content.

The authors go on to show how a model trained in such a way, which they refer
to as CTRLsum, can generate entity-focused summaries, by using an entity name
as the keyword prefix. Additionally, providing differing numbers of keywords
can be used to control the length of the generated summary.  The authors also
show that CTRLsum can respond sensibly to prompts, i.e. instead of providing
keywords, a question or initial phrase is provided.  Useful summarization
behavior can be achieved including zero shot question answering, or
enumeration of a research paper's contributions or the purpose of an
invention.

While the paper feels largely like an extension of Keskar et al. 2019, the
evaluation of the proposed methods is very thorough on a variety of settings
and domains. I especially enjoyed the break out of entity targeted summaries
based on whether the entity occurred in the lead and/or reference summary. The
use of prompts to obtain question answering and more focused
contributions/purpose summarization was also very interesting.

I would be happy to see this paper accepted to ICLR. This paper offers a
simple method of obtaining a variety of focused or targeted summarization
behaviors from a BART summarization model. In general, I would like to see
more work like this exploring methods of controlling pretrained language
models.  The evaluation of the correctness of the generated utterances
suggests that this method provides fairly reliable control.

There several areas where the paper could improve. The explanation of how
length control is achieved was not very clear. It would help to have examples
like those shown in the appendix present in the section introducing length
control.

Comparisons are to the standard BART model or to Fan et al. (2018) which
similarly prepend important control information to the input. It would be
interesting to see an evaluation that compared CTRLsum to BART with a
constrained decoding method, such as dynamic beam allocation [1].

Additionally, the authors should say more about the differences in entity
control of their method and Fan et al. 2018, which seem on their face to be
similar.

Did the authors experiment with pairs of entities as entity controls? It would
be especially interesting to see whether the model preserves the correct
relationship between entities, especially for entities that didn't occur in
the same sentences in the original document, e.g. one important entity and one
unimportant entity.

[1] Matt Post and David Vilar. Fast Lexically Constrained Decoding with Dynamic Beam Allocation for Neural Machine Translation. ACL. 2018.

---

> ### Author Response · Authors · 2020-11-24
> **Response to Reviewer #2**
>
> We are glad you like this paper and thank you for the encouraging comments!
>
> > Q1: the authors should say more about the differences in entity control of their method and Fan et al. 2018, which seem on their face to be similar.
>
> Thank you for the advice! Fan et al. and our method in entity control look similar only at test time. At training time Fan et al. require entity annotations while ours do not. Also, Fan et al. has a more sophisticated procedure to train their entity-control model as quoted from their paper: “at training time ….. To ensure the entity request is informative, we provide an entity that is present in the ground-truth summary but not present in the summary generated by the baseline model”. We have added the details about our difference into Section 4.2 Entity Control section.
>
> > Q2: Did the authors experiment with pairs of entities as entity controls? It would be especially interesting to see whether the model preserves the correct relationship between entities,
>
> This is an interesting point! We added several randomly picked qualitative examples from paired entities into Appendix E.2. Please check the updated paper for details. We use one important entity and one unimportant entity as input as suggested by the reviewer, and the two entities do not appear in the same sentence in most cases. From our perspective, the output does look interesting -- it seems like the model performs reasonably well to preserve the correct relationship between entities, the model is also able to generate the “hidden entities” correctly which are used to connect the input pair in some cases.
>
> >Q3: It would be interesting to see an evaluation that compared CTRLsum to BART with a constrained decoding method, such as dynamic beam allocation.
>
> Thank you for the advice! We agree that comparing with constrained decoding methods is very interesting. We think the two are very different approaches which might share similar performance, the constrained decoding methods don’t bother to train a new model but require a more complex and slower decoding process. Thus it is definitely worth exploring the tradeoff between these two, we don’t have the time to do it for the rebuttal but will consider adding it in the future.

---

### Official Review · AnonReviewer4 · 2020-10-28
**Interesting techniques for some summarization tasks, but unclear contribution over Fan et al.**

**Rating:** 5
**Confidence:** 5

**Review:**

# Summary:

Builds/extends on Controllable Abstractive Summarization (Fan et al) using keywords and other prompts. There’s two phases, and both phases are independent:

1. Extract keywords, z,  using a BERT classifier/sequence-tagger trained to predict keywords
2. Fine-tune BART (Lewis et al) to learn p(summary | document, z).

One can use automatic keywords using (1) and get uncontrolled generation, for which they present SOTA results on some summarization tasks. Two datasets are collected: (a) intro->contributions from arxiv papers; (b) patent->one-sentence summary, which are used to measure performance of prompts specific to those tasks.

# Pros:
1. Improves state-of-the-art results on some summarization benchmarks.
2. Provides BERTScore results in addition to ROUGE.
3. Results provided across multiple summarization datasets.
4. Interesting new datasets for measuring document+prompt->summary performance
5. Interesting zero-shot/transfer results from summarization to Question-answering.

# Cons:
1. Contribution in methods over Fan et al + BART (Lewis et al) is minimal. Results in Fan et al were weak because the underlying model was much weaker than BART, so it is unclear how much results here improved from simply using BART and adding control tokens from Fan et al, which would be a useful baseline to have that is omitted.
2. Since the focus of the paper is on controlling generation, more results on this would be informative. It is unclear how well control works when not using oracle words or automatically extracted keywords, i.e. user-controlled. An MTurk experiment evaluating how well control works would be useful in assessing this.
3. Comparing BART/PEGASUS to BART+BERT-based model is a little unfair since BERT is another large model in the system, i.e. the total amount of compute and number of parameters is much greater. A more fair comparison would be to compare using a smaller BART or PEGASUS model such that the model sizes are comparable. It is unclear whether the proposed system would do better than BART/PEGASUS scaled to the same amount of total parameters.
4. For the prompt tests, how well does BART/PEGASUS do if the decoder is prompted? This baseline would be useful to have. That is, it’s unclear how the p(y | x, z) improves over using p(y | x) by simply prompting the decoder.

# Clarifications/questions:
1. For "CONTRIBUTION AND PURPOSE SUMMARIZATION", what is the fine-tuning process? Are the models fine-tuned on (paper/patent, keywords)->abstract task before testing on intro->contribution generation?
2. How are entities randomly selected in the example decodes in the Appendix?
3. Are any of the prompts used in training or is it zero-shot?
4. What is zero-shot state-of-the-art on the QA tasks? Please add to Table 5. GPT-3 zero-shot results would also be informative.

---

> ### Author Response · Authors · 2020-11-24
> **We have added new human eval results and clarified the difference between this work and Fan et al.**
>
> We thank the reviewer for the time and comments. Due to time limitations, we could only address major points, but we’ll try to reflect all advice in future revisions.
>
>
> > Q1: For the prompt tests, how well does BART/PEGASUS do if the decoder is prompted? This baseline would be useful to have. That is, it’s unclear how the p(y | x, z) improves over using p(y | x) by simply prompting the decoder.
>
> We would like to clarify that in the submission version the BART baselines reported in all prompt tests (contribution, purpose, QA) were already prompted with the same prompts as CTRLsum, thus we believe that Table 5 and Table 6 exactly showed the improvement of p(y | x, z) over p(y | x) that simply prompts the decoder, as suggested by the reviewer. We did include this configuration detail in the submission version for contribution and purpose summarization (in both caption and model names from Table 6), now we added it to QA (in Table 5 caption) as well to make it clearer.
>
>
> > Q2: Contribution in methods over Fan et al + BART (Lewis et al)  is minimal.
>
> We would like to highlight the difference between our work and that of Fan et al.
>
> Methodologically, (1) Fan et al. require to pre-define the test control aspects before training, and cannot generalize to unseen control aspects at test time. For example, if they train a length-control system, then the model is incapable of performing entity control at test time, and vice versa. Similarly, length-control or entity-control systems from Fan et al. may not perform purpose summarization or QA tasks well after we demonstrate the necessity of *non-entity* keywords in addition to prompts. We emphasize that our *generic* model does not require a prior notion of test control aspects at training time,  but is still able to perform multi-dimensional control at test time as the five example applications in this paper.
>
> (2) Moreover, the method proposed by Fan et al. requires additional, manual annotations collection of length and entities (which may not be easy to obtain in many domains) to train a length or entity control system. In contrast, our method only uses keywords for training that are automatically extracted based on articles and reference summaries. Also, the entity-control method from Fan et al. needs another baseline model to provide additional information for training, as quoted from their paper: “at training time ….. To ensure the entity request is informative, we provide an entity that is present in the ground-truth summary but not present in the summary generated by the baseline model”.
>
> Empirically, among five diverse control aspects that we demonstrated, the only overlapping experiments between this paper and Fan. et al. are entity and length control. For length control, we implemented the approach from Fan. et al. with BART (LengthCode in Table 4) and found that it fails to control length when the underlying architecture is strong like BART, with a Pearson correlation coefficient being zero between length control code and summary length. For entity control, it is true that our success rate is much higher because of BART (in the submission version we actually ran CTRLsum with convolutional seq2seq as Fan et al. and analyzed the effect of underlying architectures in Appendix B), we have rephrased the Results paragraph in Section 4.2 as well as the captions of Table 3 to make this clearer, stating that the CTRLsum numbers are not comparable to Fan et al. and we included their numbers only for reference point. We did not re-implement their entity-control approach with BART because the two approaches are not comparable anyway due to the requirement of entity annotations from their method.
>
> Therefore, both methodologically and empirically, there are substantial differences in contributions between our work and Fan et al.
>
> > Q3: Results in Fan et al were weak because the underlying model was much weaker than BART.
>
> We believe that this concern is only regarding entity control results though the reviewer does not specify. To clarify, the length control approach from Fan et al does not work when applied with BART as we showed in Table 4. Regarding entity control, we agree with the reviewer’s point and we rephrased descriptions in Section 4.2 as mentioned in response to #Q2 above. However, we would also like to emphasize that the methods of Fan et al. and CTRLsum are not comparable for entity control even with the same architecture because (1) Fan et al. require entity annotations for training while CTRLsum does not, and (2) their method is specifically designed for entity control and unable to generalize to unseen control aspects at test time while ours aims to be generic, as explained in response to Q2.

---

> > ### Author Response · Authors · 2020-11-24
> > **Cont'd part 1**
> >
> > > Q4: It is unclear how well control works when not using oracle words or automatically extracted keywords, i.e. user-controlled. An MTurk experiment evaluating how well control works would be useful in assessing this.
> >
> > Evaluating CTRLsum as a fully interactive system with user-arbitrarily-specified keywords has many difficulties: (1) non-expert human evaluations can be unreliable and exhibit poor correlation with experts even for traditional summarization system [1,2], and how to obtain reliable human evaluation for summarization from non-expert MTurkers is still an active research problem [3], (2) when equipped with user input, human evaluation becomes even harder with more freedom to vary summaries for a single article -- the evaluation rubric becomes far more subjective due to diverse and ill-defined "user intent", (3) maybe more importantly, in the interactive case it is difficult to clearly define what aspect we are controlling and what the application scenario is, which brings difficulty to a clear problem formulation of this paper.
> >
> > We emphasize that the difficulties above are exactly the reasons that we narrow down the freedom and focus on five well-defined applications in the first place. In contrast with unclear/ill-defined "user intent" in the case of arbitrary keywords without any constraints, our experiments simulate user intent in specific and well-defined tasks such as  entity control, length control, contribution summarization, etc., and this paper aims to have a focused contribution on such well-defined application scenarios. Therefore, despite the interesting potentials of CTRLsum conditioned on unconstrained arbitrary keywords, evaluation from such perspective is beyond the scope of this paper.
> >
> > However, we do share the concerns with the reviewer on evaluating control with human evaluations. Thus we added human evaluation results for entity control and purpose summarization in Section 4.7, Table 8. The human annotators were informed with the pre-defined user intent (generating entity-focused summary or purpose-focused summary) and asked to score control directly. Please find details and results in the updated paper. We also note that we didn’t perform human eval for contribution summarization since it is difficult for the annotators to judge contributions of scientific papers from various domains.
> >
> > To summarize the results, the control accuracy and control relevance (see definition in Section 4.7) in all cases are >= 3.5 out of 5. The control accuracy of important entity control and purpose control are comparable between BART and CTRLsum without significant difference (p-value > 0.05), while CTRLsum shows significantly better control relevance overall by focusing solely on the desired information. Also, the unconstrained BART are unable to generate unimportant-entity-related summaries and thus suffers from poor scores.
> >
> > We hope these clarification and new results can address your concerns!
> >
> >
> > [1] Fabbri et al. Summeval: Re-evaluating summarization evaluation. arXiv 2020
> >
> > [2] Gillick et al. Non-expert evaluation of summarization systems is risky. NAACL 2010 Workshop on Creating Speech and Language Data with Amazon’s Mechanical Turk.
> >
> > [3] Shapira et al. Crowdsourcing Lightweight Pyramids for Manual Summary Evaluation. NAACL 2019
> >
> > > Q5: Comparing BART to BART+BERT-based models is a little unfair due to different total parameter sizes.
> >
> > We agree this for uncontrolled summarization. Our method follows previous work in hybrid, extractive- abstractive/compressive summarization where an additional network is used for the extraction stage [4,5,6,7]. Such models were also benchmarked against single-model solutions. Yet, we do share the reviewer’s concerns on this line of work, and we think the reviewer’s suggested experiment is interesting to understand the effect of overall parameter size in this case, we’ll consider running it in the future.
> >
> > We would also like to emphasize that CTRLsum does not require a BERT in most of the control experiments (entity, contribution, purpose, QA), and thus the parameter size is the same as BART in those cases.
> >
> >
> > [4] Chen et al. Fast Abstractive Summarization with Reinforce-Selected Sentence Rewriting. ACL 2018
> >
> > [5] Liu et al. Generating Wikipedia by Summarizing Long Sequences. ICLR 2018
> >
> > [6] Xu et al. Neural Extractive Text Summarization with Syntactic Compression. EMNLP 2019
> >
> > [7] Desai et al. Compressive Summarization with Plausibility and Salience Modeling. EMNLP 2020
> >
> >
> > > Q6: For "CONTRIBUTION AND PURPOSE SUMMARIZATION", what is the fine-tuning process? Are the models fine-tuned on (paper/patent, keywords)->abstract task before testing on intro->contribution generation?
> >
> > Yes, the modes are fine-tuned with (paper/patent, keywords) -> abstract. We just use the standard summarization dataset for training, and the training task/method for all datasets is always the same regardless of the test task.

---

> > > ### Author Response · Authors · 2020-11-24
> > > **Cont'd part 2**
> > >
> > > > Q7: How are entities randomly selected in the example decodes in the Appendix?
> > >
> > > We identify all named entity mentions in the test article, remove duplicate entity mentions, and randomly sample 5 from the remains without replacement following a uniform distribution.
> > >
> > > > Q8: Are any of the prompts used in training or is it zero-shot?
> > >
> > > None of the prompts are used in training and it is zero-shot. The training only uses article, summary, and keywords prepended to articles.
> > >
> > > > Q9: What is zero-shot state-of-the-art on the QA tasks? Please add to Table 5. GPT-3 zero-shot results would also be informative.
> > >
> > > As far as we know, the zero-shot state-of-the-art reading comprehension is probably the giant GPT-2/GPT-3 language models. The GPT-3 paper does not report performance on NewsQA or SQuAD v1.1, and we don’t have access to the GPT-3 model. However, we evaluated the GPT2-Large model on the two QA benchmarks and added the results to Table 5. We could not evaluate the GPT2-XL model with 1.5B parameters on our single GPU device.

---

### Official Review · AnonReviewer5 · 2020-11-06

**Rating:** 7
**Confidence:** 3

**Review:**

This paper proposes a two-stage summarization system where a document is provided along with (optionally) keywords or a prompt. This supplemental information helps to guide the summarization and possibly make it more user-specific. The keywords and prompt can also be guessed automatically by a BERT-base model, which seems to improve automatic metrics on CNN/daily mail.

Strengths:
* Moving beyond the conventional 'document/summary' framework of existing summarization approaches is a strength of this paper. This paper studies a few different ways that the summaries can be controlled: through prompts, entities, or one-sentence summaries of summaries (contribution and purpose summarization). These seem novel at least to this reviewer and could be helpful for future work.
* When using oracle guidance, performance increases on several different datasets for slightly different forms of summarization (CNN/DM, arxiv, bigpatent).
* The idea of using a two-stage approach (with a BERT-Base extractor to guess keywords to guide the summary) seems novel to this reviewer, and it seems to enable this approach to perform well even in an unconditional setting.

Weaknesses:
* The main weakness to this reviewer is that the evaluation might not be sufficiently convincing to test the key hypothesis: that these keywords/prompts can enable users to get summaries that are closer to their intent (like Figure 1). To this reviewer, this necessitates a human evaluation. Though testing factual correctness in Table 3 seems like a good start to this reviewer, measuring overall summarization quality (both conditional and unconditional on user intent) through a human evaluation seems necessary.
* (minor) one possible reason why the two-stage approach might perform better on unconditional summarization is because there are more parameters when ensembling BERT-Base and BART. Possibly doing something multitask within a single BART model might be cleaner and could clearly test whether the gains come from more parameters/computation, or the keyword approach.


Overall, to this reviewer, this paper seems like it would be strong if it had human evaluations of summarization quality. I would be willing to raise my score if those were provided.

----

Update: thanks for the additional human evaluation results! These help and the results on excluding unimportant entities seem strong to this reviewer. Perhaps it might be more helpful for the annotators themselves to try to interact with the summarizer in some way, but that's a more minor point.

Anyways, I bumped up my score from 5->7.

---

> ### Author Response · Authors · 2020-11-24
> **We have added new results on human evaluation in Section 4.7.**
>
> We thank the reviewer for the time and comments. Due to time limitations, we could only address major points, but we will try to reflect all advice in future revisions.
>
> > Q1: More human evaluation of the overall summarization quality
>
> Thank you for the advice! We have added a new Section 4.7 to the paper to include more human evaluation results on both controlled and uncontrolled summarization. While the reviewer suggests evaluating “overall summarization quality”, we argue that more detailed evaluation over multiple dimensions can be more reliable, informative, and objective [1] (for example, some annotators may simply prefer more fluent and coherent summaries even though its factual correctness is worse than others if we only ask for an overall score). Therefore, we follow previous work [2,3] and ask the human annotators to score multiple dimensions of the summary such as factual consistency, relevance, fluency, and coherence. Please find details and results in the updated paper. Here we summarize the key findings:
>
> (1) For uncontrolled summarization, the quality of summaries from all systems across different datasets and dimensions is generally good, as most of the scores are higher than 4.0 out of 5.
>
> (2) For uncontrolled summarization, our significance test results suggest that both BART and CTRLsum (oracle) are *not* significantly different from CTRLsum (automatic), despite very different similarities against reference summaries in terms of ROUGE/BERTScore (e.g. CTRLsum with oracle keywords has much higher scores on automatic metrics than others).  This implies that the summary quality from different systems powered by strong pretrained models like BART has become difficult to be clearly distinguished by non-expert MTurkers. More expertise might be needed to pursue more reliable human evaluation for such strong systems. The reliability of crowdsourcing workers have been studied in [3,4], and how to obtain more reliable results from non-expert annotators for summarization is still an open research problem [1].
>
> (3) For controlled summarization, the control accuracy and control relevance (see definition in Section 4.7) are >= 3.5 out of 5 in all cases. The control accuracy for important entity control and purpose control are comparable between BART and CTRLsum without significant difference (p-value > 0.05), while CTRLsum shows significantly better control relevance overall by focusing solely on the "user intent". Also, the unconstrained BART are unable to generate unimportant-entity-related summaries and thus suffers from poor scores.
>
> We would like to note that we didn’t perform human eval for contribution summarization since it is difficult for the annotators to judge contributions of scientific papers from various domains. We hope these new results can address your concerns!
> .
>
> [1] Shapira et al. Crowdsourcing Lightweight Pyramids for Manual Summary Evaluation. NAACL 2019
>
> [2] Grusky et al. Newsroom: A Dataset of 1.3 Million Summaries with Diverse Extractive Strategies. NAACL 2018
>
> [3] Fabbri et al. Summeval: Re-evaluating summarization evaluation. arXiv 2020
>
> [4] Gillick et al. Non-expert evaluation of summarization systems is risky. NAACL 2010 Workshop on Creating Speech and Language Data with Amazon’s Mechanical Turk.
>
>
> > Q2: The two-stage approach might perform better because there are more parameters when assembling BERT and BART.
>
> We do agree that the two-stage model in sum has more parameters and can be a bit unfair for (uncontrolled summarization) comparison. Our method follows previous work in hybrid, extractive- abstractive/compressive summarization where an additional network is used for the extraction stage [5,6,7,8]. Such models were also benchmarked against single-model solutions. We share the reviewer’s concerns on this line of work, and we think the reviewer's suggestion of using the same BART to perform both tagging and summarization with multi-task learning is a novel idea worth being explored in the future.
>
> We would also like to emphasize that CTRLsum does not require a BERT in most of the control experiments (entity, contribution, purpose, QA), and thus the parameter size is the same as BART in those cases.
>
>
> [5] Chen et al. Fast Abstractive Summarization with Reinforce-Selected Sentence Rewriting. ACL 2018
>
> [6] Liu et al. Generating Wikipedia by Summarizing Long Sequences. ICLR 2018
>
> [7] Xu et al. Neural Extractive Text Summarization with Syntactic Compression. EMNLP 2019
>
> [8] Desai et al. Compressive Summarization with Plausibility and Salience Modeling. EMNLP 2020

---

### Author Response · Authors · 2020-11-24
**Revision Submitted**

We thank all the reviewers for their helpful comments! We have submitted a revised manuscript and made the following modifications to address the reviewers' major concerns:

-- Added a human evaluation section (Section 4.7) which includes new human evaluation results for both controlled and uncontrolled summarization across multiple tasks and datasets (Reviewer #4, Reviewer #5).

-- Added ablation results (Appendix C, Table 11) including CTRLsum (keyword only), CTRLsum (prompt only), and CTRLsum (keyword + prompt) on different control tasks (Reviewer #1).

-- Added qualitative examples (Appendix E.2)  with paired entity control (Reviewer #2)

-- Clarified several statements in the paper

While limited by time in the response period, we do still plan to address all the reviewer’s comments in future revisions. We also welcome any further feedbacks to improve this paper !

---

### Decision · Program_Chairs · 2021-01-07
**Final Decision**

**Decision:**

Reject

**Comment:**

 The paper attempts at controllable summarization in two dimensions: Length, and content. Authors try to achieve this through training data generation approach, where they provide a standard BART model with additional keywords (extracted using a BERT model) in training.

The paper's main motivation on controllable summarization is important and interesting, and despite simplicity, the results are generally positive on multiple datasets.
However, despite positive results, reviewers raised several critical concerns, some of which remained unresolved after reviewer/author discussion period. Examples include concerns regarding lack of methodological novelty over prior work (R1, R2, R4), unfair/incomplete comparisons with prior work (R2, R4, R5), and not evaluating on a real user controlled setting instead of automatic keywords (R1, R4). Although the authors tried addressing human evaluation in their revision, some reviewers remained unconvinced.
Some quotes from reviewer discussions:

> I'm not convinced the human eval was done properly.

> My concerns are not completely addressed and the score remains unchanged. For human evaluation, I agreed with Reviewer X.